# Beyond Numeric Rewards: In-Context Dueling Bandits with LLM Agents

## Abstract

In-context reinforcement learning (ICRL) is a frontier paradigm for solving reinforcement learning problems in the foundation model era. While ICRL capabilities have been demonstrated in transformers through task-specific training, the potential of Large Language Models (LLMs) out-of-the-box remains largely unexplored. Recent findings highlight that LLMs often face challenges when dealing with numerical contexts, and limited attention has been paid to evaluating their performance through preference feedback generated by the environment. This paper is the first to investigate the performance of LLMs as in-context decision-makers in the problem of Dueling Bandits (DB), a stateless preference-based reinforcement learning setting that extends the classic Multi-Armed Bandit (MAB) model by querying for preference feedback. We compare GPT-3.5 Turbo, GPT-4, GPT-4 Turbo, Llama 3.1, and o1-preview against nine well-established DB algorithms. Our results reveal that our top-performing LLM, GPT-4 Turbo, possesses an understanding of relative decision-making sufficient to achieve low weak regret in DB by quickly selecting the best arm in duels. However, we observed that an optimality gap exists between LLMs and classic DB algorithms in terms of strong regret. LLMs struggle to converge and consistently exploit even when explicitly prompted to do so, and are sensitive to prompt variations. To overcome these issues, we introduce an agentic flow framework: **L**LM with **E**nhanced **A**lgorithmic **D**ueling (**LEAD**), which integrates off-the-shelf DB algorithms with LLM agents through fine-grained adaptive interplay. We show that LEAD has theoretical guarantees inherited from classic DB algorithms on both weak and strong regret. We validate its efficacy and robustness even with noisy and adversarial prompts. The design of such an agentic framework sheds light on how to enhance the trustworthiness of general-purpose LLMs used for in-context decision-making tasks.

## 1 Introduction

Large sequence models pretrained on offline reinforcement learning datasets have led to the emergence of in-context reinforcement learning (ICRL) (Laskin et al., 2022; Lee et al., 2024), where models can infer tasks from interaction histories as context and make effective decisions in unseen environments without parameter updates. Through trial and error, these models can self-improve their policies purely in-context. While ICRL capabilities have been demonstrated in transformers with task-specific training from scratch, the potential of general-purpose Large Language Models (LLMs) to perform ICRL remains largely unexplored. Recent investigations into LLMs' out-of-the-box ICRL capabilities in environments with numeric rewards have reported notable failure cases, e.g., LLM agents being vulnerable to adversarial loss functions and suffering from high regret compared to classic algorithms such as Follow-The-Regularized-Leader (FTRL) (Park et al., 2024), and exhibiting failures in exploration within Multi-Armed Bandit (MAB) problems (Lattimore & Szepesvári, 2020) via standard training (Krishnamurthy et al., 2024). Even with inference-time algorithmic guidance, an optimality gap persists between LLMs and classic (contextual) MAB algorithms (Nie et al., 2024). These results suggest that carefully designed prompts and non-trivial algorithmic interventions are needed to elicit desirable in-context reinforcement learning behavior of LLM agents.

The failure cases encountered by LLMs may be attributed to intrinsic difficulties in processing numeric rewards, especially in tasks where patterns are difficult to express in natural language. Recent findings have pointed out that LLMs often struggle with simple numerical comparisons (e.g., incorrectly judging 13.11 to be larger than 13.8), and there has been a notable lack of emphasis

Figure 1: In-context reinforcement learning of an LLM agent with numeric rewards (in a multi-armed bandit environment) and preference feedback (in a dueling bandit environment).

on evaluating the relative comparisons among the decisions they generate. Figure 1 shows a toy example illustrating the in-context interaction between an LLM agent and different environment settings. To disentangle the complexities introduced by numerical rewards, this paper focuses on the problem of Dueling Bandits (DB) (Yue et al., 2012; Zoghi et al., 2014b), a stateless preference-based reinforcement learning setting (Wirth et al., 2017; Pacchiano et al., 2021) that extends the classic MAB model by querying for *preference feedback* between selected pairs of arms to identify the best one. In DB, the agent learns through binary outcome (win or lose) of a noisy comparison between the two selected arms. This setup is particularly useful when eliciting explicit feedback is challenging or when the feedback is inherently comparative, like taste of food and product attractiveness (Yue et al., 2012). DB has attracted significant attention due to its applicability in information retrieval (Yue & Joachims, 2009), recommendation systems (Sui et al., 2017), and online ranker evaluation (Zoghi et al., 2014b). We frame our investigation with the following question:

*Are LLMs effective in-context agents for solving the problem of dueling bandits?*

The DB problem poses distinctive challenges as a relative decision-making instance, particularly due to the sparse nature of the relative rewards. This sparsity complicates the in-context decision-making process, as it restricts the feedback obtained from interactions, introducing a level of difficulty not typically seen in conventional bandit problems. Even though reduction from DB to standard MAB exists (Ailon et al., 2014; Saha & Gaillard, 2022), it remains unclear how LLMs would perform in DB with preference feedback rather than numeric rewards. There are conceptual differences between them, similar to those between Reinforcement Learning from Human Feedback (RLHF) (Stiennon et al., 2020) and standard RL, where impossibility results can be found in (Wang et al., 2024b).

While task-specific training of large sequence models can yield promising ICRL results, it is often impractical due to the substantial computational resources required. Similar to the settings in (Mirchandani et al., 2023; Krishnamurthy et al., 2024; Nie et al., 2024; Chen et al., 2024), we evaluate the emergent zero-shot abilities (Wei et al., 2022) of ICRL in general-purpose LLMs under the dueling bandit problem, without re-training or fine-tuning. We summarize our main results below.

**Evaluation of LLMs' Emergent Zero-Shot Abilities of In-Context Dueling Bandits.** We go beyond numeric rewards to evaluate the performance of LLM agents in terms of both strong and weak regret for making decisions in DB by comparing against various baseline DB algorithms via a case study. We found that the top-performing general-purpose LLMs possess an understanding of relative decision-making sufficient to achieve low weak regret in DB, which significantly differs from that in classic MAB settings (Krishnamurthy et al., 2024). Notably, GPT-4 TURBO can serve as an effective decision-maker for dueling bandits in the short term, quickly selecting the best arm in duels with low variance across a range of instances. However, consistent with (Nie et al., 2024), we found that an optimality gap exists between LLMs and classic DB algorithms in terms of strong regret. The long-term performance is hindered by over-estimation bias in the exploration stage and lack of convergence criterion in the exploitation stage. This highlights the need for more effective and robust strategies to bridge this gap for in-context dueling bandits.

**Effective and Robust Agentic Flow Framework for In-Context Dueling Bandits.** To address the identified optimality gap and enhance the trustworthiness of in-context LLM agents in DB tasks, in Section 4.1, we propose an agentic flow framework, **L**LM with **E**nhanced **A**lgorithmic **D**ueling (LEAD) that integrates off-the-shelf Explore-then-Exploit DB algorithms with LLM agents. This framework enables the fine-grained adaptive interplay between rule-based expert systems and in-context LLM agents, enhancing their ability to handle DB problems via algorithmic interventions as suggested in (Krishnamurthy et al., 2024; Nie et al., 2024). As an illustrative example, we demonstrate how Interleaved Filter2 (IF2) algorithm can be incorporated with LLM agents in this framework. We show that the proposed framework has theoretical guarantees, with experiments demonstrating its efficacy and robustness across various prompting scenarios.

## 2 PRELIMINARIES

In this section, we briefly introduce the problem of dueling bandits (DB) and establish the necessary notation for this paper. Additional useful definitions can be found in Appendix B.3.1.

**Dueling Bandits.** In a fundamental $K$-armed dueling bandit problem setting (Yue et al., 2012), a learner interacts with the environment by selecting two arms $\mathsf{Arm}_1(t)$ and $\mathsf{Arm}_2(t)$ from a set of $K$ arms $\{b_1, \ldots, b_K\}$ for a noisy comparison (a duel), at each round $t \in \{1, \ldots, T\}$ as Figure 1 illustrates. The outcome of a duel between two arms $(i, j)$ is probabilistic. More precisely, the event that an arm $b_i$ wins against $b_j$ is a Bernoulli random variable with a parameter denoted by $\Pr(b_i \succ b_j)$. For notational convenience, we normalize $\Pr(b_i \succ b_j)$ such that $\Pr(b_i \succ b_j) = \epsilon(b_i, b_j) + 1/2$, where $\epsilon_{ij} := \epsilon(b_i, b_j) \in (-1/2, 1/2)$ is a measure of the *distinguishability* between arms $b_i$ and $b_j$, which is stationary over time and is symmetric such that $\epsilon_{ij} = -\epsilon_{ji}$ for all $i, j \in [K] := \{1, \ldots, K\}$. Finally, for notational convenience, we define a preference matrix $P = [\epsilon_{ij}]_{i,j \in [K]}$.

**In-Context LLM Agents for Dueling Bandits.** We consider an LLM agent with policy $\pi_{\mathrm{LLM}}$ interacting with a $K$-armed dueling bandit environment in-context. At each round $t \in \{1, \ldots, T\}$, the LLM agent selects a pair of arms $(\mathsf{Arm}_1(t), \mathsf{Arm}_2(t))$ from the set $\{b_1, \ldots, b_K\}$ based on a natural language instruction $\mathtt{Prompt}(C, H_t, R)$ (see Figure 7), consisting of three parts:

- Problem Description $P$: a natural language description of the DB problem, including the number of arms $K$, the time horizon $T$, and the task objective.
- History $H_t$: an externally summarized interaction history (Krishnamurthy et al., 2024) up to round $t$, which includes a sequence of pairwise dueling results and the empirical probabilities.
- Reasoning $R$: the zero-shot chain-of-thought (CoT) reasoning (Kojima et al., 2022) that encourages the LLM agent to reason about the problem in a structured manner.

The LLM agent's policy can be represented as:

$$(\mathsf{Arm}_1(t), \mathsf{Arm}_2(t)) = \pi_{\mathrm{LLM}}\left(\mathtt{Prompt}(P, H_t, R)\right). \tag{1}$$

The goal is to maximize the cumulative reward over some time horizon $T$, where the reward is the sum of the unknown probabilities of the two chosen arms beating the best arm (Condorcet winner). We can quantify performance as minimizing the cumulative regret, either in the strong or weak sense (see Eq.(4) and Eq.(5)).

**Strong and Weak Regret.** Throughout this paper, we assume the standard setting that a *Condorcet winner* (CW) exists (Sui et al., 2017; Wu & Liu, 2016; Zoghi et al., 2014b; Yue et al., 2012). The CW denoted as $b^*$ is an arm that is preferred over all the other arms, i.e., $b^* = b_i$ if $\epsilon_{ij} > 1/2$ for all $j \in [K] \backslash \{i\}$. We consider two performance metrics: (i) *strong regret* (SR), which evaluates the total preference gap between $b^*$ and both selected arms; (ii) *weak regret* (WR), which compares $b^*$ only with the better of the two arms. Detailed definitions and settings are provided in Appendix B.3.1.

**Related Works.** Our work contributes to the growing community of intersection between LLMs and decision-making. We summarize the detailed related works about dueling bandits, LLM agents for bandits, and LLMs for in-context decision-making in the Appendix A.

## 3 LLMS AS STANDALONE IN-CONTEXT DECISION-MAKERS

To evaluate the LLMs' efficacy for solving DB problems in-context, in this section, we use LLMs as standalone decision-making agents and compare them with classic DB algorithms. Our evaluation is two-fold: First, in Figures 2 and 9, we compare the performance of LLMs and classic algorithms in terms of the strong and weak regret (see Eq.(4) and Eq.(5)), with standard deviation. Second, we delve into the experimental results and analyze the success and failure modes of LLM agents.

### 3.1 IMPLEMENTATION DETAILS OF EXPERIMENTS

**Prompts and Configurations of LLMs.** We employ an interactive zero-shot chain-of-thought (CoT) prompt $\mathtt{Prompt}(P, H_t, R)$, as defined in Section 2, which describes the problem setting $P$ along with an externally summarized interaction history $H_t$ and reasoning instructions $R$. We adopt the prompting template and LLM configurations that lead to the best performance among all prompt variations explored in a recent study (Krishnamurthy et al., 2024) for the classic MAB problem. The

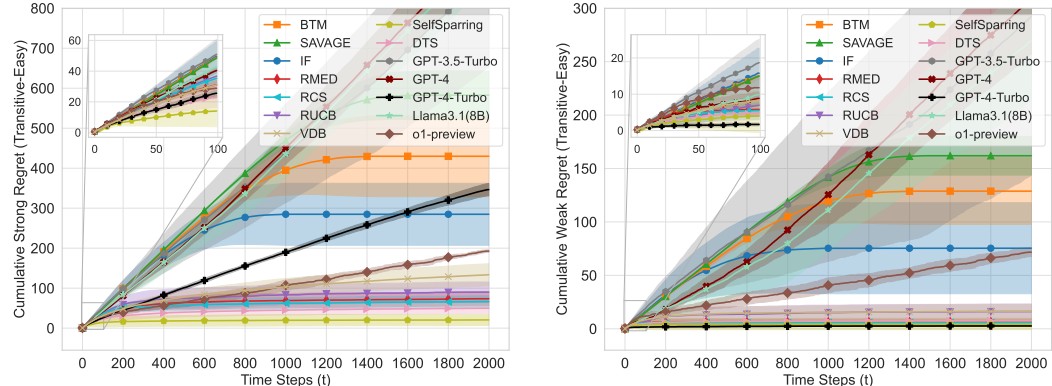

Figure 2: Comparisons between LLM agents and DB algorithms. **Left** and **Right**: strong and weak regret for the `Transitive-Easy` instance. Results for `Transitive-Hard` are in Figure 9.

LLM agents interact with dueling bandit environments in a round-based manner, with the prompt guiding their decision-making process. We conduct experiments with **five** LLMs: GPT-3.5 TURBO, GPT-4, GPT-4 TURBO, LLAMA 3.1, and O1-PREVIEW. Note that we skip the GPT-4O version which is primarily developed for multimodal tasks and has the same intelligence as GPT-4 TURBO. The detailed prompt is provided in Appendix C.1.2.

**Baselines.** We compare LLMs against **nine** well-established baseline algorithms to evaluate their efficacy. The baselines include Interleaved Filter (IF2) (Yue et al., 2012), Beat the Mean (BTM) (Yue & Joachims, 2011), Sensitivity Analysis of VAriables for Generic Exploration (SAVAGE) (Urvoy et al., 2013), Relative Upper Confidence Bound (RUCB) (Zoghi et al., 2014a), Relative Confidence Sampling (RCS) (Zoghi et al., 2014b), Relative Minimum Empirical Divergence (RMED) (Komiyama et al., 2015), Versatile Dueling Bandits (VDB) (Saha & Gaillard, 2022), Self-Sparring (Sui et al., 2017), and Double Thompson Sampling (DTS) (Wu & Liu, 2016). Each of these algorithms employs distinct strategies for selecting arms and estimating preferences, with the ultimate goal of efficiently identifying the Condorcet winner. We assess the performance of LLMs and baseline algorithms using strong regret and weak regret metrics defined in Section 2.

**Environments.** We evaluate the regret performance of LLMs and baselines across two types of stochastic environments under the standard DB setting with a Condorcet winner (CW). The environments differ in their stochastic transitivity properties and are divided into two cases, each with two levels of difficulty instances (Easy and Hard) depending on the distinguishability of the CW in beating other arms: (i) Transitive case (SST ∩ STI): This case uses a Bradley-Terry-Luce (BTL) model (Bradley & Terry, 1952; Yue et al., 2012). The preference matrices generated in this way satisfy the Strong Stochastic Transitivity (SST) and Stochastic Triangle Inequality (STI), which implies the existence of a CW; (ii) Intransitive case (CW \ (SST ∪ STI)): the preference matrices introduce cyclic preferences among non-winning arms while ensuring the existence of a CW. The intransitive case is modeled using a custom preference construction designed to violate SST and STI. The detailed constructions can be found in Appendix C.1.1.

**Random Tests.** The scale of our experiments is chosen to balance computational feasibility while preserving the ability of obtaining meaningful conclusions. We set the time horizon to $T = 2000$ rounds, providing the LLMs and baseline algorithms with sufficient opportunity to learn and adapt to the DB environments. Each experiment is replicated $N = 5$ times for the LLMs and $N = 20$ times for the baseline algorithms, enabling an understanding of their average behaviors and reliable performance estimates.

## 3.2 EXPERIMENTAL RESULTS

For brevity, we present our initial analysis focused on the `Transitive-Easy` instance (Figure 2). The analysis is qualitatively similar for the `Transitive-Hard` instance (Figure 9 in Appendix C.2). We use $\gamma = 0.5$ for BTM, $f(K) = 0.3K^{1.01}$ for RMED, $\eta = 1$ for Self-Sparring, and $\alpha = 0.51$ for RUCB, RCS and DTS. We analyze the results in terms of the strong and weak regret defined in Section 2. In the following sections, we will mainly focus on GPT-4 TURBO, which is our top-performing LLM, highlighting its success and failure modes.

**Emergence of in-context dueling bandits abilities.** While GPT-3.5 TURBO and GPT-4 fail to solve the DB problem, GPT-4 TURBO consistently outperforms state-of-the-art (SOTA) DB baselines in weak regret (see Figures 2 and 9). This reveals that the in-context dueling bandits abilities emerge as the general capabilities grow via standard training methods in general-purpose LLMs. Figure 13 (left?) illustrates the fraction of duels including the best arm across different time intervals. GPT-4 TURBO outperforms other LLMs and the DB baselines throughout the entire timeline. These findings suggest that GPT-4 TURBO can effectively process the preference feedback obtained from duels and make informed decisions to quickly identify and include the best arm in its duels.

**Stable performance across different instances.** GPT-4 TURBO demonstrates low variance compared to other LLMs and baselines across varying levels of difficulty. As shown in Figure 14, GPT-4 TURBO exhibits the lowest average generalized variance of strong and weak regret in both instances. This highlights its ability to maintain a stable decision-making process in DB.

> **Best Arm Identification:** LLMs' in-context dueling bandits abilities emerge as the general capabilities grow. The Condorcet Winner is consistently selected in duel via GPT-4 TURBO, leading to exceptional weak regret performance with minimal variance.

**Exploration Vulnerability.** In the exploration stage, we observe that GPT-4 TURBO tends to quickly narrow down to a small subset of arms (although usually containing the Condorcet winner) and repeatedly compare these arms. In contrast, the baselines exhibit more diverse and extended exploration patterns. This behavior suggests that GPT-4 TURBO may overestimate the quality of arms that win their initial comparisons based on limited historical data. Unlike the baselines, which have explicit exploration mechanisms, LLMs rely on their inherent randomness (via sampling from their output distribution) to explore. Based on these observations, we hypothesize that if GPT-4 TURBO happens to sample a sequence of comparisons that favors suboptimal arms early on, it can get stuck comparing these arms indefinitely. To test this hypothesis, we conducted experiments using noisy prompts with biased history. Our results in Figure 16 confirm that GPT-4 TURBO's exploration strategy is indeed vulnerable to biased history initialization and can converge to local optima.

**Exploitation Inability.** Despite GPT-4 TURBO's outstanding weak regret performance, it fails to consistently converge to a single best arm to duel against itself, even when the prompt setting explicitly calls for it. This behavior highlights a fundamental limitation of LLMs: they are primarily designed and trained for word token prediction rather than decision-making. Unlike baselines with explicit stopping conditions, GPT-4 TURBO relies on its inherent language modeling capabilities to determine when to stop exploring. Consequently, in the later exploitation stage, GPT-4 TURBO keeps comparing the same top arms without committing to a single winner (see Figure 3). This suggests that the language modeling objective alone may not be sufficient for LLMs to achieve optimal control in complex decision-making tasks like DB.

> **Lack of Robust Strategy:** LLMs' performance can be hindered by overestimation bias in the exploration stage and the lack of convergence criteria in the exploitation stage.

**Biased understanding of DB problem during pretraining.** Our two best-performing LLMs, GPT-4 TURBO and O1-PREVIEW, exhibit systematic biases regarding the DB problem, likely due to a lack of exposure to similar tasks during pretraining. Specifically, they incorrectly assume that an arm cannot duel with itself (the convergence case), even when explicitly prompted to do so (see examples in Appendix C.1.3). This misunderstanding makes the DB problem as an out-of-distribution (OOD) task for LLMs, and in-context instructions fail to fully override this internal bias. Consequently, LLM agents cannot completely align with problem descriptions due to the inherent limitations of in-context learning, which cannot really generalize to OOD tasks (Wang et al., 2024a). Figure 13 supports these observations: O1-PREVIEW demonstrates better reasoning capabilities by transitioning from exploration to exploitation effectively and achieving lower strong regret than GPT-4 TURBO. However, its CoT mechanism reinforces its internal biased understanding of DB, resulting in poorer weak regret performance due to the selection of two suboptimal arms in duels.

> **Systematic Biases:** LLMs out-of-the-box lack a fundamental understanding of the DB problem and instead intuitively choose the next pair of arms to compare based on dueling history.

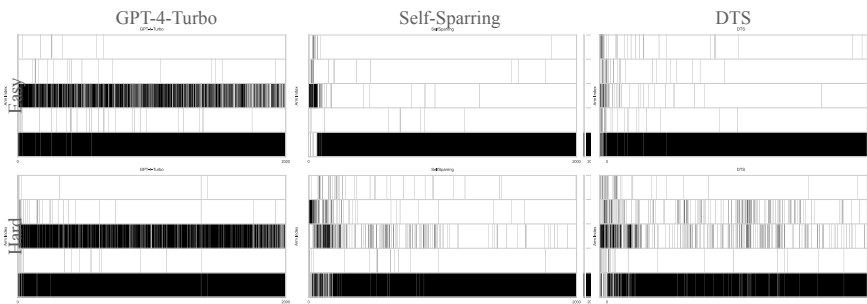

Figure 3: Comparison of duel selection trajectories among GPT-4 TURBO, SELF-SPARRING, and DTS on the `Transitive-Easy` (Top Row) and `Transitive-Hard` (Bottom Row) instances. The decision trajectories of GPT-4 TURBO exhibit a clear pattern of continuous exploration without converging to the best arm. In contrast, SELF-SPARRING and DTS demonstrate more structured exploration patterns and convergence properties on both instances.

**Scalability Limitation.** To evaluate whether LLMs can generalize their exceptional weak regret performance, we conducted experiments from two perspectives: (i) Removing transitivity in preference structures: we change from transitive cases to intransitive cases that violate SST and STI (see Figures 10 and 11). The analysis of `Transitive-Easy` and `Transitive-Hard` is qualitatively similar: LLMs fail to replicate their weak regret performance in transitive cases when faced with intransitive instances. This finding suggests that while LLMs exhibit emergent abilities for relative decision-making rooted in linguistic knowledge, their effectiveness is constrained to specific transitive scenarios. In DB settings with intransitive preference structures, their weak regret performance is no longer exceptional; (ii) Increasing the number of arms: as illustrated in Figure 12, from $K = 5$ to $K = 10$, GPT-4 TURBO's performance exhibits a noticeable decline with the increase in $K$. The finding suggests that when faced with a larger number of arms, LLMs struggle to effectively infer the relative strengths among them. To quantify this scalability limitation and formally characterize the dueling bandit instances LLMs can handle, we introduce the concept of the **Relative Decision Boundary (RDB)**. The RDB for a given LLM $m$ is defined as the set of problem difficulties $\mathcal{D}$ for which the model achieves an acceptable level of weak regret, satisfying the condition:

$$\text{RDB}(m) = \left\{ (K, \mathcal{T}, \Delta) \,\middle|\, \text{WR}(m, \mathcal{D}(K, \mathcal{T}, \Delta)) \leq \text{R}_{\text{th}} \right\}. \tag{2}$$

Here, $\text{WR}(m, \mathcal{D})$ represents the cumulative weak regret incurred by the model $m$ on a problem of difficulty $\mathcal{D}$, while $\text{R}_{\text{th}}$ is a predefined threshold that quantifies acceptable weak regret performance. Overall, RDB is influenced by the inherent ability of $m$, the number of arms $K$, the transitivity $\mathcal{T}$, the distinguishability between arms $\Delta$.

> **Fail to generalize:** LLMs' performance degrades when introducing intransitive preference structures or large number of arms. We introduce Relative Decision Boundary (RDB) to describe the dueling bandit instances that LLMs can effectively handle.

After characterizing in-context LLM agents in DB, to further investigate the algorithmic behavior of LLMs and develop more robust and effective in-context decision-making strategies, we seek to answer the following questions:

**[Q1]** *Can we develop an Algorithm-Enhanced in-context DB agent with a theoretical guarantee?*

**[Q2]** *How does it perform compared to standalone LLM agents and classic DB algorithms?*

## 4 ALGORITHM-ENHANCED LLMS FOR DUELING BANDITS

Classic DB algorithms based on the Explore-then-Exploit framework, such as Interleaved Filter 2 (IF2) (Yue et al., 2012), are known to be near-optimal, with matching regret upper and lower bounds up to multiplicative constants. To address the challenges identified in Section 3.2 of using standalone LLM agents for DB, we propose an algorithm-enhanced approach: **L**LM with **E**nhanced **A**lgorithmic **D**ueling (**LEAD**) to demonstrate the possibility of integrating off-the-shelf DB algorithms with LLM agents through fine-grained adaptive interplay. Our framework, LEAD, enjoys both a regret guarantee and strong empirical performance.

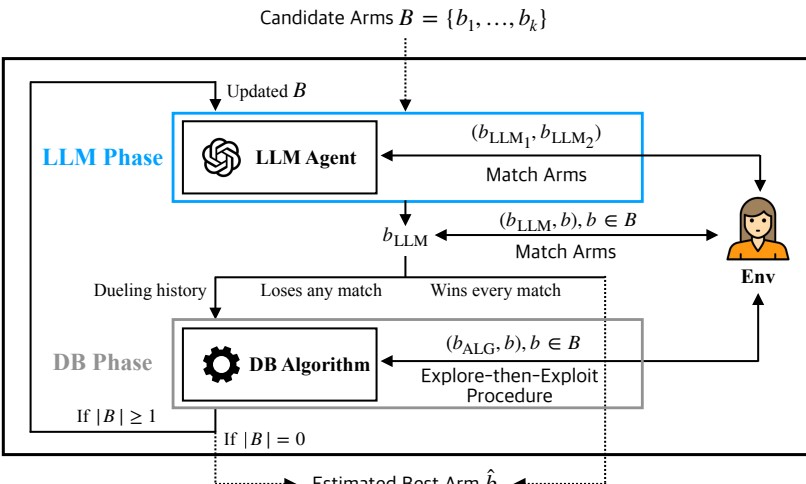

Figure 4: Main components of the proposed LEAD agent in Algorithm 1 are illustrated: (i) The blue-colored part represents the **LLM phase**. (ii) The grey-colored part indicates the **DB phase**. (iii) The Algorithmic Procedures are detailed in Appendix B.2. (iv) The black arrows denote shared interactions between components. (v) The dotted arrows represent the input and output.

## 4.1 ALGORITHMIC DESIGN OF LEAD

In this section, we present the design intuitions of LEAD. We begin by discussing the limitations of a naive intervention approach and the desirable properties for an effective Algorithm-Enhanced LLM framework. Based on these considerations, we propose an agentic framework design LEAD, where we can incorporate any Explore-then-Exploit DB algorithms (Zoghi et al., 2014b). As an illustrative example, we use IF2 (Yue et al., 2012) to demonstrate how off-the-shelf algorithms can be integrated within LEAD and provide a detailed description.

**Limitations of Naive Intervention.** A straightforward approach to addressing the convergence instability limitation of LLMs is to use a simple if-else condition that forces the LLMs to converge when they first exploit two identical arms, which we call the Convergence-Triggered (CT) intervention strategy. However, CT fails to guarantee the selection of the true Condorcet winner and can reinforce local optima (see Figure 17 in Appendix C.2 for a failure example). This suggests that relying on the LLMs' internal convergence behavior to trigger the transition from exploration to exploitation is unreliable, as the LLMs are largely driven by its inherent sampling noise rather than a structured exploration policy. Thus, handling this limitation with theoretical guarantees remains challenging.

**Desirable Properties for LLM Augmentation.** To address **[Q1]**, we seek an algorithmic framework with the following properties: (i) A clear, symbolic logical structure that allows for easy integration with LLM & Algorithm suggestions; (ii) A well-defined exploration-exploitation trade-off that leverages the LLMs' exploration behavior while ensuring convergence; (iii) Strong theoretical guarantees to maintain robustness with various prompting scenarios.

As a result, we find that the Explore-Then-Exploit structure is particularly well-suited for LLMs (see Appendix B.1 for a detailed illustration). By selecting an Explore-Then-Exploit DB algorithm as the foundation inside LEAD, we address **[Q1]**. As an example, we use IF2 as the base to illustrate the theoretical guarantee and empirical performance. This approach can be applied similarly to other algorithms with regret bounds in the Explore-Then-Exploit family.

**Algorithmic Framework.** The procedures of the LEAD are illustrated in Figure 4 and presented in Algorithm 1 (see more details in Appendix B.2). LEAD (IF2 base) maintains a confidence parameter $\delta$ and a threshold parameter $\epsilon$ that control the algorithm's confidence of matches between arms. The key components of LEAD (IF2 base) are as follows:

- **Phase 1 (LLM Phase):** *Utilization of LLM recommended arms*: The agentic framework maintains a set of candidate arms $B$. Given two arms suggested by an LLM agent, the framework begins with finding a winner between them, denoted by $b_{\text{LLM}}$. The winning arm $b_{\text{LLM}}$ is then matched with each remaining arm $b \in B$. This phase continues until $b_{\text{LLM}}$ is defeated or all arms in $B$ have been matched. The variable TrustLLM is used to control the execution of the LLM phase, and it is set to

---

**Algorithm 1** Algorithm-Enhanced LLM Agent: LEAD (IF2 base)

---

**Initialize :** Time horizon length $T$, arms $B = \{b_1, \ldots, b_K\}$, incumbent arm $b_{\text{IF2}}$

1  **while** $|B| \geq 1$ **do**
2  | TrustLLM $\leftarrow$ True                           /* **LLM Phase** in Figure 4 (Lines 2-10) */
3  | **while** TrustLLM **do**
4  | | Prompt LLM to select $(b_{\text{LLM}_1}, b_{\text{LLM}_2})$ from $B$
5  | | $b_{\text{LLM}} \leftarrow$ MATCH ARMS$(b_{\text{LLM}_1}, b_{\text{LLM}_2})$ (Procedure 1)           /* Compare LLM arms */
6  | | **for** $b \in B$ **do**
7  | | | $b' \leftarrow$ MATCH ARMS$(b_{\text{LLM}}, b)$ (Procedure 1)    /* Compare $b_{\text{LLM}}$ with others */
8  | | | **if** $b' \neq b_{\text{LLM}}$ **then** TrustLLM $\leftarrow$ False, **continue**
9  | | **end**
10 | **end**
11 | StillTrust, $B \leftarrow$ VALIDATE$(b', B, \text{TrustLLM})$ (Procedure 2)
12 | $b_{\text{IF2}}, B \leftarrow$ IF2$(b_{\text{IF2}}, B)$ (Procedure 3)        /* **IF2 Phase** in Figure 4 (Lines 11-12) */
13 **end**
14 **if** StillTrust **then return** $b_{\text{LLM}}$
15 **else return** $b_{\text{IF2}}$

---

False when $b_{\text{LLM}}$ is defeated by another arm, indicating that the LLM's suggestions are no longer trusted.

- **Phase 2 (IF2 Phase):** *Roll back to* IF2: If $b_{\text{LLM}}$ is defeated, the framework switches to implementing one round of IF2 with an incumbent arm $b_{\text{IF2}}$ selected based on an estimated preference matrix $\hat{P}$.

After **Phase 2**, the algorithm-enhanced agent repeats **Phase 1** until $B$ only contains the best arm. Algorithm 1 and Figure 4 summarize the phases above, with details delegated to Appendix B.2.

## 4.2 THEORETICAL GUARANTEES FOR LEAD

In this section, we begin by characterizing the vulnerability of using standalone LLM agents for dueling bandits in Theorem 4.1. Then, we provide the theoretical guarantees of LEAD in Theorem 4.2 and 4.3, demonstrating its efficacy and convergence.

**Theorem 4.1** (Vulnerability). *For the dueling bandits problem with $K$ arms and time horizon $T$, there exists a preference structure and an attacker strategy with budget $\Phi(T)$, such that any standalone LLM agent, whose policy is represented by Eq.(1) and whose worst-case behavior under the original prompt satisfying Assumption 4, will suffer an expected regret of $\Omega\left(\min\left\{\Phi(T), T/K\right\}\right)$.*

The proof of Theorem 4.1 is provided in Appendix B.3.2. The theorem underscores the suboptimality of standalone LLM agents in DB problems, particularly when input prompts are subjected to adversarial attacks. This vulnerability highlights the need for a more robust approach to use in-context LLM agents while offering theoretical guarantees under diverse prompting scenarios.

**Regret Bounds.** Following the algorithmic design of LEAD in Section 4.1, LEAD (IF2 base) inherits the theoretical guarantees of IF2 (see Appendix B.3.1), while nontrivially leveraging the benefits of LLMs' exceptional weak regret performance for exploration across a range of instances within RDB. Specifically, LEAD (IF2 base) has the following theoretical guarantee:

**Theorem 4.2** (Expected Regret). *Suppose for $t \geq T_{\text{LLM}}$, the arms recommended by an LLM agent contain the best arm $b^*$. Under Assumptions 1-3, the expected strong regret of LEAD (IF2 base) satisfies $\mathbb{E}\left[\text{SR(LEAD)}\right] \leq \widetilde{O}\left((K \log T)/\epsilon_{1,2}\right)$, and the expected weak regret can be bounded by*

$$\mathbb{E}\left[\text{WR(LEAD)}\right] \leq \min\left\{\widetilde{O}\left(T_{\text{LLM}} + \frac{K \log K}{\epsilon_{1,2}}\right), \widetilde{O}\left(\frac{K \log T}{\epsilon_{1,2}}\right)\right\}, \tag{3}$$

*where $\widetilde{O}(\cdot)$ hides poly-logarithmic factors of $T$.*

Note that Theorem 4.2 is general such that we do not assume any specific adversarial behaviors of the LLM agent, including Assumption 4. The proof of Theorem 4.2 is provided in Appendix B.3.2. The required assumptions are precisely stated in Appendix B.3.1. Theorem 4.2 establishes a best-of-both-worlds result in terms of the efficacy and robustness of LEAD.

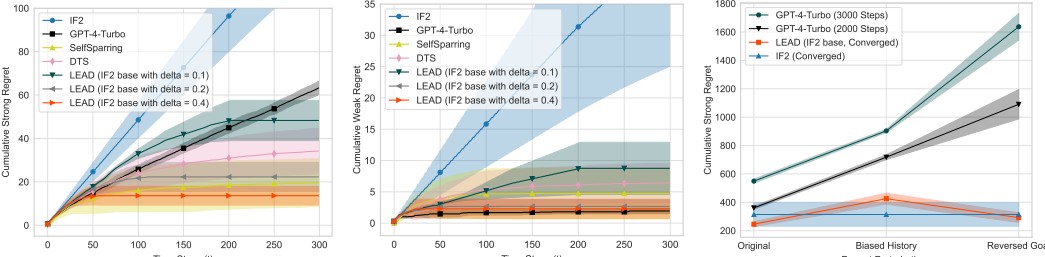

Figure 5: Comparisons between LEAD, GPT-4 TURBO, and baseline algorithms (IF2, SELF-SPARRING and DTS). **Left** and **Middle**: strong and weak regret on the `Transitive-Easy` instance. **Right**: robustness evaluation under prompt perturbations (prompts are in Appendix C.1.2).

**Efficacy.** As illustrated in Figures 2, 3, and 9, LEAD has the potential to identify the best arm after a short exploration stage. This results in strong and weak regret bounds of $\widetilde{O}(T_{\text{LLM}} + (K/\epsilon_{1,2}) \log K)$ and $O(T_{\text{LLM}})$, respectively, that are independent of the horizon length $T$, provided the LLM agent suggests a pair of arms that includes the best arm $b^*$. Furthermore, when the prompt contains extra textual context that can infer the relative preferences between arms, $T_{\text{LLM}}$ will become smaller, further enhancing the best-case performance. We consider it an important direction for future work within the Contextual Dueling Bandit framework Dudík et al. (2015).

**Guaranteed Convergence.** Additionally, both the strong and weak regret for LEAD are guaranteed to satisfy a worst-case upper bound of $\widetilde{O}\left((K/\epsilon_{1,2}) \log T\right)$, which is only worse than the information-theoretic lower bound of $\Omega\left((K/\epsilon_{1,2}) \log T\right)$ in (Yue et al., 2012) by a poly-logarithmic factor of $T$. The worst-case upper bounds on the strong and weak regret hold regardless of the specific prompting scenario, ensuring that LEAD maintains its theoretical guarantees even in the presence of noisy or adversarial prompts, as considered in Theorem 4.1. This safety guarantee is particularly important in practical applications, where the prompts provided to the LLM agent may not always be optimal.

The following theorem indicates that the additional term $(K \log K)/\epsilon_{1,2}$ in equation 3 is almost tight. Its proof is provided in Appendix B.3.2.

**Theorem 4.3** (Converse). *Given any algorithm* ALG *for dueling bandits provided with an in-context LLM agent recommending arms, if it satisfies* $\mathbb{E}\left[\text{WR}(\text{ALG})\right] \leq T_{\text{LLM}}$ *for all* $T_{\text{LLM}}$, *then it must hold* $\mathbb{E}\left[\text{SR}(\text{ALG})\right] \geq \mathbb{E}\left[\text{WR}(\text{ALG})\right] \geq \Omega\left(T\right)$ *for some instance of the LLM agent.*

### 4.3 EMPIRICAL EVALUATION OF LEAD

Regarding **[Q2]**, we design a two-fold evaluation to assess efficacy and robustness. The evaluation is conducted on the `Transitive-Easy` instance, which provides higher distinguishability, allowing us to observe convergence and regret differences within a practical number of steps. First, we compare the strong and weak regret of LEAD against state-of-the-art baseline algorithms to validate its efficacy. Second, we investigate the robustness of LEAD with noisy and adversarial prompts.

#### 4.3.1 EFFICACY EVALUATION: STRONG REGRET AND WEAK REGRET

**Hyper-parameters.** In our implementation of LEAD (see Algorithm 1), there are two hyper-parameters: the threshold parameter $t$, which controls the maximum number of comparisons between arms, and the confidence parameter $\delta$, which determines the confidence level for pruning suboptimal arms. For the threshold parameter $t$, we considered values from the set $\{50, 100, 200\}$, and for the confidence parameter $\delta$, we explored values from $\{0.1, 0.2, 0.4\}$. After fine-tuning, we found that setting $t = 50$ and $\delta = 0.4$ provided the best performance in terms of cumulative regret.

We evaluate the cumulative strong and weak regret performance of the proposed LEAD with different confidence parameter settings ($\delta = 0.1, 0.2, 0.4$) and $t = 50$: Figure 5 (Left and Middle) demonstrates that LEAD exhibits competitive performance across different $\delta$ values. For strong regret, $\delta = 0.1$ results in more conservative exploration, leading to slightly higher regret compared to baselines. As $\delta$ increases ($\delta = 0.2$ or $0.4$), LEAD achieves lower cumulative strong regret, outperforming all the baselines at $\delta = 0.4$ due to more aggressive exploration to identify the optimal arm sooner. Similarly, for weak regret, LEAD consistently achieves superior performance. When $\delta = 0.2$ and $\delta = 0.4$, LEAD effectively identifies and includes the optimal arm in comparisons. These hyper-parameter values strike a balance between the number of comparisons required to identify the best arm and the

confidence level for pruning suboptimal arms, enabling LEAD to efficiently explore and exploit the available arms in-context for the dueling bandits setting.

### 4.3.2 ROBUSTNESS EVALUATION: NOISY AND ADVERSARIAL PROMPTS

Recent studies (Loya et al., 2023; Krishnamurthy et al., 2024) have emphasized the importance of varying prompts to elicit the desired behavior from LLMs in decision-making tasks, highlighting the potential limitations of prompt quality. Results obtained from a single prompt template may lead to unreliable conclusions that cannot generalize to real-world situations where optimal prompts are often unavailable. Thus, we evaluate the robustness of LEAD by employing two types of prompt perturbations (see Figure 8) along with the original prompt (see Figure 7). Across all scenarios, LEAD demonstrates superior performance and robustness compared to standalone GPT-4 TURBO.

**Original Prompt.** Under the initial prompt, LEAD leverages the LLM's ability to quickly identify the best arm through exploration (under the DB instance within RDB Eq.(2)). As shown in Figure 6 (Top Row), we observe that LEAD benefits from the LLM's exploration ability by initializing with the best arm as the incumbent when entering the IF2 phase. Compared to GPT-4 TURBO, convergence to the Condorcet winner is guaranteed for LEAD with high probability.

**Biased History.** We inject an incorrect history into the prompt, where each non-optimal arm initially wins against the best arm 10 times, while keeping the underlying preference matrix unchanged. LLM agents are observed to get trapped in local optima for extended periods, where LEAD overcomes this limitation by employing uniform comparisons in the IF2 phase to escape such suboptimal exploration modes.

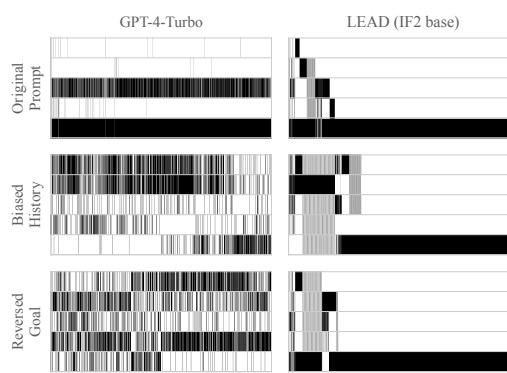

Figure 6: Duel selection trajectory of GPT-4 TURBO and LEAD under different prompt settings (Figures 7 and 8). **Top**: original prompt. **Middle**: noisy prompt (biased history). **Bottom**: adversarial prompt (reversed goal).

**Reversed Goal.** When the prompt is adversarially modified from maximizing reward to minimizing, the LLM consistently recommends non-optimal arms after its exploration stage. Even with adversarial prompts, LEAD still achieves near-optimal cumulative strong regret. Since the LLM's exploration capability is only utilized within the bounded length of the MATCH ARMS procedure, the impact of the reversed goal on the exploitation phase is mitigated.

Figure 5 (right) presents the cumulative strong regret results comparing LEAD against standalone LLM agents and the IF2 algorithm across three prompt designs. Notably, LEAD with $\delta = 1/(TK^2)$ (consistent with IF2 to showcase the robust behavior) achieves near-optimal cumulative regret with low variance even with noisy and adversarial prompts, validating the regret bounds stated in Theorem 4.2. LEAD and IF2 converge to the best arm within 2000 steps, while GPT-4 TURBO's cumulative expected regret continues to increase, indicating the instability of standalone in-context LLM agents.

## 5 CONCLUSION

This paper evaluates LLMs as in-context decision-makers for standard context-free dueling bandits (DB) with a Condorcet Winner, offering the first systematic insights into their strengths and limitations. While GPT-4 TURBO achieves exceptional weak regret performance across a range of instances, it fundamentally lacks an understanding of the DB problem. Our findings reveal that LLMs' decision-making in DB, driven by linguistic priors, lacks the necessary criteria for convergence and generalization to complex scenarios, leading to an optimality gap between LLMs and classic DB algorithms in terms of strong regret. To bridge this gap, we propose LEAD, an agentic flow framework that integrates off-the-shelf DB algorithms with LLM agents through fine-grained adaptive interplay. This framework provides theoretical guarantees and demonstrates robust performance even under noisy and adversarial prompts. Our work contributes to the problem of In-Context Reinforcement Learning (ICRL). The framework we proposed sheds light on how language-based reasoning can inspire robust frameworks that translate words into actions, paving the way for more trustworthy AI systems through the interplay between rule-based experts and in-context LLM agents.

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

## APPENDIX

This appendix provides supplementary information and additional experimental results to support the main text. The content is organized into three main parts:

**A. Related Works**

**B. Theoretical Part: Algorithm Design and Analysis of LEAD**

- Appendix B.1 presents the algorithm design logic using Explore-then-Exploit methods.
- Appendix B.2 describes the LEAD algorithm stated in Section 4.1, detailing its key features and implementation remarks.
- Appendix B.3.1 presents the necessary definitions, assumptions and lemmas for the theoretical analysis of LEAD in Section 4.2.
- Appendix B.3.2 proves Theorem 4.1, 4.2, and 4.3, establishing LEAD's regret bounds.

**C. Experimental Part: Prompt Design and Supplementary Results**

- Appendix C.1.1 illustrates the transitive and intransitive environments construction.
- Appendix C.1.2 illustrates the prompt design and prompt perturbations logic.
- Appendix C.1.3 provides exemplars of GPT-4 TURBO to showcase their behavior.
- Appendix C.2 presents supplementary experimental results, providing further insights into the performance and behavior of the algorithms in Sections 3 and 4.

## A    RELATED WORKS

We provide the detailed related works as follows.

**Dueling Bandits.** The problem of dueling bandits was initially introduced in (Yue et al., 2012). Various methods have been proposed to tackle the task since then. These methods can be broadly classified into two categories as Explore-Then-Exploit methods and Ongoing Regret Minimization methods according to (Zoghi et al., 2014b). Explore-Then-Exploit methods focus on identifying the best arm with high confidence before exploiting it, such as Interleaved Filter (IF) (Yue et al., 2012) and Beat the Mean (BTM) (Yue & Joachims, 2011), etc. In contrast, Ongoing Regret Minimization methods explicitly target the objective of minimizing cumulative regret, including Relative Upper Confidence Bound (RUCB) (Zoghi et al., 2014a) and Self-Sparring (Sui et al., 2017), etc. Dueling bandit problem and preference feedback in general has a wide variety of applications, including recommendation systems (Yue et al., 2012), robotics (Tucker et al., 2020), and most recently, the training algorithm of large language models, such as Reinforcement Learning from Human Feedback (RLHF) (Ouyang et al., 2022).

**LLM Agents for Multi-Armed Bandits.** Several recent works have explored evaluating the capabilities of LLMs in bandit problems. For example, (Baheri & Alm, 2023) proposed an approach to enhance contextual bandits by integrating LLMs as encoders. The LLMs' ability to capture rich semantic and syntactic information from textual contexts is leveraged to provide the algorithm with a more informative representation of the context. The LLM-augmented algorithm transforms the raw context into a latent space vector using the LLM's encoding capabilities. This encoded context is then used to guide the decision-making process. (Krishnamurthy et al., 2024) investigates whether LLMs can engage in exploration in simple MAB environments without additional training. They compared various prompt designs and found that GPT-4 with zero-shot chain-of-thought (CoT) reasoning and an externally summarized interaction history performed the best, while other configurations failed in exploration, either by never selecting the best arm after initial rounds or by selecting all arms nearly equally often. Different from the previous results, in this work we go beyond the settings of numeric rewards and investigate the capabilities of LLMs under preference feedback.

**In-Context LLMs for Decision-Making.** Beyond bandit problems, LLM agents have demonstrated strong capabilities in complex reasoning across a wide range of in-context reinforcement learning and decision-making tasks (Laskin et al., 2022; Lee et al., 2024; Zhou et al., 2022; Yao et al., 2024). Various existing works aim to understand LLM agents' capabilities for in-context decision-making, with notable examples including planning (Huang et al., 2022; Hao et al., 2023). Additionally, LLM agents have been shown to enhance embodied agents in robotic applications by providing advanced

task planning abilities (Brohan et al., 2023) and reward designing (Ma et al., 2023), further enabling the development of lifelong learning agents (Wang et al., 2023). Besides these empirical successes, the authors of (Park et al., 2024) analyzed LLMs' interactions in online learning and game theory settings through the lens of the regret metrics. They identified simple cases where LLMs fail to be no-regret. Another line of research incorporates LLMs into classic decision-making frameworks to create LLM-augmented online decision-makers. For instance, Liu et al. (Liu et al., 2024) utilized LLMs to enhance the components of warm starting, sampling candidates, and surrogate modeling in Bayesian optimization. Our work contributes to this broad area by integrating LLM agents with the classic Explore-then-Exploit DB algorithms to enhance the utilization of preference feedback.

# B  ALGORITHM DESIGN AND ANALYSIS OF LEAD

In this section, we detail the design principles and implementation of the LEAD algorithm. First, we present the algorithm design logic. Then, we provide a rigorous proof of Theorem 4.1, 4.2, and 4.3, establishing the theoretical guarantees of LEAD (IF2 base) under the assumptions outlined in Appendix B.3.1.

## B.1  ALGORITHM DESIGN LOGIC

**Explore-then-Exploit algorithms as Ideal Candidates.** Classic DB algorithms can be classified into two categories: Explore-Then-Exploit methods and Ongoing Regret Minimization methods (Zoghi et al., 2014b). Among these, Explore-Then-Exploit structure stands out as particularly well-suited for LLM augmentation:

- The Explore-Then-Exploit structure naturally aligns with the LLMs' tendency to keep exploring without converging (see Figure 3), allowing for leveraging the LLMs' exploration behavior while mitigating their exploration vulnerability and convergence instability (see Table **??**).

- Its symbolic representation of the algorithm's logic enables clear integration of LLM suggestions at specific points without disrupting the overall structure and theoretical guarantees. In contrast, algorithms like Self-Sparring in (Sui et al., 2017) are less symbolic, making them less suitable for direct LLM augmentation.

- Its strong theoretical guarantees, e.g., IF2 with an expected regret bound of $O((K/\epsilon_{\text{bad}}) \log T)$ matching the DB problem's lower bound of $\Omega((K/\epsilon_{\text{bad}}) \log T)$ up to constants (see Appendix B.3.1), and its empirical performance (see Figures 2 and 9) provide a robust foundation, ensuring convergence and bounded regret.

## B.2  DETAILED PROCEDURE DESCRIPTION

In Procedure 1 below, we describe the MATCH ARMS procedure used in LEAD (see Algorithm 1 and Figure 4).

---

**Procedure 1** MATCH ARMS (with a bounded number of comparisons)

**Input**: Two arms $a, a'$, confidence parameter $\delta \leftarrow 1/(K^2 \log T)$, and threshold $\epsilon \leftarrow \epsilon_{1,2}$

**if** $a \neq a'$ and $t \leq (16/\epsilon^2) \log(K \log T)$ **then**

    **while** $\nexists \, (b, b') \in B$ such that $\hat{P}_{b,b'} > 1/2$ and $1/2 \notin \hat{C}_{b,b'}$ **do**

        Compare $a$ with $a'$ and update $\hat{P}_{a,a'}$ and $\hat{C}_{a,a'}$, $t \leftarrow t + 1$

    **end**

    **return** $b$

**else return** $a$

---

We reprise the IF2 procedure in (Yue et al., 2012) below to complement the presentation of LEAD.

It is worth noting the following features of Algorithm 1 in its practical implementation.

**Remark 1.** *The LLM Phase allows for flexible exploration design within the bounded length of the* MATCH ARMS *procedure, not limiting the number of prompts and comparisons performed by the LLM to identify an empirically best arm.*

---

**Procedure 2** VALIDATE

**Input**: Incumbent arm $a$, candidate arms $B$, TrustLLM, confidence parameter $\delta \leftarrow 1/(TK^2)$, and threshold $\epsilon \leftarrow \epsilon_{1,2}$

**if** TrustLLM *is* True **then**
 **for** $b \in B$ **do**
  **if** $t \leq (16/\epsilon^2) \log(K \log T)$ **then**
   **while** $\nexists\, (b, b') \in B$ *such that* $\hat{P}_{b,b'} > 1/2$ *and* $1/2 \notin \hat{C}_{b,b'}$ **do**
    Compare $a$ with $b$ and update $\hat{P}_{a,b}$ and $\hat{C}_{a,b}$, $t \leftarrow t + 1$
   **end**
  **if** $b \neq a$ **then return** StillTrust $\leftarrow$ False, $B \leftarrow B \backslash \{a\}$
 **end**
 **return** StillTrust $\leftarrow$ True, $B \leftarrow \emptyset$
**if** TrustLLM *is* False **then return** StillTrust $\leftarrow$ False, $B \leftarrow B$

---

**Procedure 3** IF2 PROCEDURE

**Input**: Incumbent arm $a$, candidate arms $B$, confidence parameter $\delta \leftarrow 1/(TK^2)$, $t \leftarrow 0$

**if** $t \leq (16K/\epsilon_{1,2}^2) \log(K \log T)$ **then**
 **for** $b \in B$ **do**
  Compare $a$ with $b$ and update $\hat{P}_{a,b}$ and $\hat{C}_{a,b}$, $t \leftarrow t + 1$
 **end**
 $a, B \leftarrow$ ANNEAL$(a, B)$
**return** $a, B$

---

**Procedure 4** ANNEAL

**Input**: Incumbent arm $a$, candidate arms $B$, confidence parameter $\delta \leftarrow 1/(TK^2)$, matrices $\hat{P}$ and $\hat{C}$

**while** $\exists\, (b, b') \in B$ *such that* $\hat{P}_{b,b'} > 1/2$ *and* $1/2 \notin \hat{C}_{b,b'}$ **do**
 $B \leftarrow B \backslash \{b'\}$
**end**
**if** $\exists\, b' \in B$ *such that* $\hat{P}_{a,b'} < 1/2$ *and* $1/2 \notin \hat{C}_{a,b'}$ **then**
 **while** $\exists b \in B$ *such that* $\hat{P}_{a,b} > 1/2$ **do**
  $B \leftarrow B \backslash \{b\}$            /* IF2 pruning */
 **end**
 $a \leftarrow b'$, $B \leftarrow B \backslash \{b'\}$
**return** $a, B$

---

**Remark 2.** *The bound length in the* MATCH ARMS *procedure can be adjusted based on empirical requirements. Modifying the confidence parameter $\delta$ and the threshold $\epsilon$ will affect the regret bound and the algorithm's performance. These parameters can be tuned to balance exploration and exploitation, depending on the specific application and desired level of confidence.*

### B.3 THEORETICAL ANALYSIS

#### B.3.1 USEFUL ASSUMPTIONS AND LEMMAS FOR DUELING BANDITS

We introduce the useful definitions, assumptions and lemmas for Dueling Bandits that are necessary for the theoretical analysis of our proposed algorithm.

Throughout this paper, we consider two important performance metrics. The first is the *strong regret* of a given algorithm ALG, defined as

$$\mathsf{SR}(\mathrm{ALG}) := \sum_{t=1}^{T} \Big( \epsilon \left( b^*, \mathsf{Arm}_1(t) \right) + \epsilon \left( b^*, \mathsf{Arm}_2(t) \right) \Big). \tag{4}$$

where $T$ is the time horizon. The second is the *weak regret* of ALG, defined as

$$\mathsf{WR}(\mathrm{ALG}) := \sum_{t=1}^{T} \min \Big( \epsilon \left( b^*, \mathsf{Arm}_1(t) \right), \epsilon \left( b^*, \mathsf{Arm}_2(t) \right) \Big). \tag{5}$$

which only compares $b^*$ against the better of the two selected arms $\mathsf{Arm}_1(t)$ and $\mathsf{Arm}_2(t)$. It is worth highlighting that LLM agents exhibit significantly different behaviors with respect to the two defined notions of regret, as detailed in Section 3.2.

**Assumption 1** (Total Ordering). *The preference matrix $P = (\epsilon_{ij})$ satisfies the Total Ordering (TO) property such that for all $i, j \in [K]$, $i \succ j$ implies $\epsilon_{ij} > 1/2$.*

With the TO property satisfied, we assume the preference matrix $P$ further satisfies the following two standard properties (Yue & Joachims, 2009; 2011; Yue et al., 2012).

**Assumption 2** (Strong Stochastic Transitivity). *The preference matrix $P = (\epsilon_{ij})$ satisfies the Strong Stochastic Transitivity (SST) such that for any arms $i, j, k \in [K]$ such that $i \succ j \succ k$ under the total order $\succ$, we have $\epsilon_{ik} > \max\{\epsilon_{ij}, \epsilon_{jk}\}$.*

**Assumption 3** (Stochastic Triangle Inequality). *The preference matrix $P = (\epsilon_{ij})$ satisfies the Stochastic Triangle Inequality (STI) such that for any arms $i \succ j \succ k$, we have $\epsilon_{ik} \leq \epsilon_{ij} + \epsilon_{jk}$.*

Note that the Bradley-Terry-Luce (BTL) model (Bradley & Terry, 1952) used in our experiments 3.1 satisfies Assumption 2 and 3. We restate the following theoretical guarantees for IF2 that is useful in the proof of Theorem 4.2. Let $\epsilon_{\text{bad}} := \min_{b \neq b^*} \epsilon(b, b^*)$.

**Lemma 1** (Theorem 2 in (Yue et al., 2012)). *Assuming the preference matrix $P$ satisfies the SST and STI, then IF2 has its expected regret (both weak and strong) bounded from above by*

$$\mathbb{E}[\mathsf{SR}(\text{IF2})] \leq O\left(\frac{K}{\epsilon_{\text{bad}}} \log T\right). \tag{6}$$

The following expected regret bound achieved by IF2 is tight up to multiplicative constants, as indicated by the lower bound (Theorem 4) in (Yue et al., 2012) such that any algorithm ALG for DB satisfies $\mathbb{E}[\mathsf{SR}(\text{ALG})] = \Omega\left((K/\epsilon_{\text{bad}}) \log T\right)$.

### B.3.2 THEORETICAL GUARANTEES OF LEAD

**Part I: Vulnerability of Standalone LLM Agents**

**Assumption 4** (Worst-Case Behavior). *Under the original prompt (see Figure 7), the worst-case behavior of an LLM agent in the dueling bandit setting is equivalent to a randomizer that selects action pairs uniformly at random.*

**Vulnerability of Standalone LLM Agents.** Inspired by the adversarial corruptions framework introduced in (Hajiesmaili et al., 2020) for the classic MAB problem, we investigate the vulnerability of standalone LLM agents in the DB setting under adversarial prompts. We consider an attacker with a budget $\Phi(T)$ who employs the following strategy: whenever the LLM agent selects the optimal arm $b^*$ for comparison, the attacker manipulates the input prompt to the LLM to eliminate $b^*$ from the duel with probability $p$ (where $0 < p \leq 1$ is a constant), subject to the constraint of performing at most $\Phi(T)$ attacks over $T$ rounds. This adversarial strategy compels the LLM agent to select suboptimal arms, resulting in poor performance, as formalized in the following theorem with Assumption 4.

***Proof of Theorem 4.1.*** Consider the following DB instance with $K \geq 3$ arms $\{b_1, \ldots, b_K\}$ and preference matrix $P$:

$$P_{i,j} = \begin{cases} 0.5 + \epsilon, & \text{if } b_i = b^* \text{ and } b_j \neq b^*, \\ 0.5 - \epsilon, & \text{if } b_i \neq b^* \text{ and } b_j = b^*, \\ 0.5, & \text{otherwise.} \end{cases}$$

where $0 < \epsilon < 0.5$ is a small constant. In this instance, arm $b^*$ is the unique Condorcet winner, and all other arms are suboptimal with a gap of $\epsilon$ to $b^*$.

Now, consider an attacker strategy with budget $\Phi(T)$: Whenever the LLM agent selects a duel containing the optimal arm $b^*$, the attacker manipulates the input prompt to the LLM agent (as described in Eq. (1)) to eliminate $b^*$ from the comparison with probability $p$ (where $0 < p \leq 1$ is a constant), subject to the constraint that the attacker can perform at most $\Phi(T)$ attacks over the $T$ rounds.

Let $N(T)$ be the number of rounds in which the LLM agent selects a duel containing the optimal arm $b^*$ up to round $T$. Due to the attacker's manipulation of the input prompt, in each of these $N(T)$ rounds, $b^*$ is eliminated from the comparison with probability $p$. However, because of the attacker's budget constraint, the actual number of attacked rounds is at most $\min\{N(T), \Phi(T)\}$.

In the rounds where $b^*$ is eliminated from the comparison, the LLM agent can only select from the suboptimal arms $\{b_i \mid b_i \neq b^*, i \in [K]\}$. Let $\Delta_i = P_{b^*, b_i} - 0.5$ denote the suboptimality gap of arm $b_i$ with respect to $b^*$. Then, the expected regret incurred in each round where $b^*$ is eliminated from the comparison is at least $\min_{b_i \neq b^*} \Delta_i = \epsilon$.

Thus, the expected cumulative regret of the LLM agent after $T$ rounds is at least:

$$\mathbb{E}[\text{Regret}(T)] \geq p \cdot \mathbb{E}[\min\{N(T), \Phi(T)\}] \cdot \epsilon \geq p \cdot \min\{\mathbb{E}[N(T)], \Phi(T)\} \cdot \epsilon,$$

where the first inequality follows from the regret incurred in rounds where $b^*$ is eliminated from the duel, and the second inequality holds due to Jensen's inequality and the linearity of expectation.

According to the Assumption 4, in the worst case, the LLM agent's behavior is equivalent to randomly selecting a duel in each round. For $K$ arms, there are $K(K-1)/2$ possible duel combinations. Therefore, the probability of selecting a duel containing $b^*$ in each round is $(K-1)/\binom{K}{2} = \frac{2}{K}$, which yields $\mathbb{E}[N(T)] = T \cdot \frac{2}{K}$. The regret bound becomes:

$$\mathbb{E}[\text{Regret}(T)] \geq p \cdot \min\left\{\frac{2T}{K}, \Phi(T)\right\} \cdot \epsilon = \Omega\left(\min\left\{\frac{T}{K}, \Phi(T)\right\}\right).$$

Therefore, any standalone LLM agent whose policy is represented by Eq. (1) under the worst-case assumption will suffer an expected regret of $\Omega\left(\min\left\{\Phi(T), \frac{T}{K}\right\}\right)$. This lower bound demonstrates the vulnerability of solely relying on LLM agents for DB in adversarial environments when the attacker can manipulate the input prompts. $\qquad\square$

**Part II: Expected Regret Bounds of LEAD (IF2 base)**

Suppose at each step $t \leq T$, aligning with the design of IF2 in (Yue et al., 2012), $\hat{P}_t$ is estimated such that each $\hat{P}_{i,j}$ is the fraction of number of comparisons when $b_i$ was the winner out of all previous $t$ comparisons. Define a confidence interval $\hat{C}_t := (\hat{P}_t - c_t, \hat{P}_t + c_t)$ where $c_t := \sqrt{\log(1/\delta)/t}$. Before proceeding to prove Theorem 4.2, we first state a useful lemma from (Yue et al., 2012) as a result of the Hoeffding's inequality (Hoeffding, 1994).

**Lemma 2** (Generalized Lemma 1 in (Yue et al., 2012)). *Let $\delta = 1/(K \log T)^2$ be a confidence parameter with $\delta \in (0, 1/2]$, a winner between two arms $b_i$ and $b_j$ is identified with probability at least $1 - \delta$, using at most $\left(16/\epsilon_{i,j}^2\right) \log(K \log T)$ number of comparisons.*

Note that Lemma 2 can be directly implied by Lemma 1 in (Yue et al., 2012). Now, under Assumption 2 and 3 such that the preference matrix $P$ satisfies the SST and STI properties, we prove Theorem 4.2.

***Proof of Theorem 4.2***. Suppose the arms suggested by LLM agent includes the best arm $b^*$ after exploring $T_{\text{LLM}}$ steps. We prove the two bounds shown in Theorem 4.2 one-by-one.

**Weak Regret Bound.** The first $T_{\text{LLM}}$ steps induce accumulated weak regret of at most $O(T_{\text{LLM}})$. According to (Yue et al., 2012), IF2 plays $O(K)$ matches (comparisons) in expectation. Thus, the expected number of rounds of calling IF2 PROCEDURE is $O(\log T / \log(K \log T))$. Applying Lemma 2, with $O\left((1/\epsilon_{1,2}^2) \log(K \log T)\right)$ (by setting a hyper-parameter $\epsilon = \epsilon_{1,2}$) comparisons between two arms, since the best arm $b^*$ is always included in each comparison, the best arm $b^*$ is correctly identified with probability at least $1 - 1/(K \log T)^2$. This procedure leads to no weak regret since $b^*$ suggested by the LLM agent is always included as the incumbent arm in future comparisons.

Moreover, the implementation of Procedure 3 induces at most $O((K/\epsilon_{1,2}^2) \log(K \log T))$ comparisons. The validation procedure (Procedure 2) leads to no weak regret if $b_{\text{LLM}}$ is indeed the best arm and the identification of $b_{\text{LLM}} = b^*$ succeeds with a probability $1 - 1/T$. Denote by $\mathcal{E}_1$ and $\mathcal{E}_2$ two error events when $b^*$ loses some of the matches in the LLM Phase. there exist comparisons (matches) fail in the validation procedure (Procedure 2) or the IF2 Phase (Procedure 3). The union bound implies with probability $1 - 1/(K \log T)$, $b^*$ will win all the matches such that $P(\mathcal{E}_1) \leq 1/(K \log T)$.

Similarly, $P(\mathcal{E}_2) \leq 1/T$. Combining these events, regarding the total expected weak regret, the expected weak regret induced by the steps after time $T_{\text{LLM}}$ can be bounded by

$$\mathbb{E}[\text{SR}(\text{LEAD after } T_{\text{LLM}})]$$

$$\leq \left(1 - \frac{1}{K\log T} - \frac{1}{T}\right) O\underbrace{\left(\frac{K\log(K\log T)}{\epsilon_{1,2}}\right)}_{\text{LLM Phase}} + \frac{1}{K\log T} O\underbrace{\left(\frac{K}{\epsilon_{1,2}}\log T\right)}_{\text{IF2 Phase}} + \frac{1}{T}\underbrace{O(T)}_{\text{Failure Cases}}$$

$$= \widetilde{O}\left(\frac{K\log K}{\epsilon_{1,2}}\right)$$

since there are at most $K + 1$ matches.

**Convergence Guarantee.** Furthermore, consider the adversarial selection of arms from the LLM agent. According to Lemma 2, the IF2 procedure with an expected regret $O\left((K/\epsilon_{1,2})\log(T)\right)$ is implemented at most $O(1)$ times with probability $1 - 1/(TK)$, provided with $|B| = K$. Therefore, the expected regret (either strong or weak) induced by each implementation of Procedure 3 is at most $O\left((K/\epsilon_{1,2})\log(T)\right)$ since there are at most $O\left((K/\epsilon_{1,2}^2)\log(K\log T)\right)$ additional comparisons of pairs in the LLM phase. Finally, applying the expected regret bound in Lemma 1 completes the proof.

$\square$

**Part III: Converse**

In the following, we argue that for any algorithm ALG, achieving an upper bound $\mathbb{E}[\text{WR}(\text{ALG})] \leq T_{\text{LLM}}$ for all $T_{\text{LLM}}$ is impossible.

***Proof of Theorem 4.3.*** Suppose ALG is an algorithm that leads to a weak regret bound $\mathbb{E}[\text{WR}(\text{ALG})] \leq T_{\text{LLM}}$ for all $T_{\text{LLM}}$, then it has to trust and include the recommended arm in all the comparisons immediately after it is proposed by the LLM agent to ensure that future weak regret becomes zero. To see this, note that one can always construct an adversarial $T_{\text{LLM}}$ that leads to a nonzero future weak regret. However, the LLM agent can choose to provide an arm that is always not the best arm for all $t \in \{1, \ldots, T\}$. This leads to $\mathbb{E}[\text{SR}(\text{ALG})] \geq \mathbb{E}[\text{WR}(\text{ALG})] \geq \Omega(T)$.

$\square$

## C PROMPT DESIGN AND SUPPLEMENTARY RESULTS

### C.1 LLM EXPERIMENTAL RESULTS

In this section, we provide the detailed design of the prompts used in our experiments and provide additional results to support our findings. We begin by presenting the original prompt used in the LLM-Env interaction and introduce the perturbed prompts, which include both noisy and adversarial variations to test the robustness of our approach. Finally, we provide four exemplars using the original prompt to to showcase the behavior of both GPT-4 TURBO and O1-PREVIEW.

#### C.1.1 ENVIRONMENTS

In transitive instances, the preference matrices are constructed using the Bradley-Terry-Luce (BTL) model (Bradley & Terry, 1952; Yue et al., 2012), with a generalized form known as the Plackett-Luce model (Plackett, 1975). In this model, each arm is associated with a utility parameter $\theta(i) > 0$, where $i$ represents the rank of the arm (i.e., $\theta(1)$ corresponds to the best arm, $\theta(2)$ corresponds to the second best arm, and so on). For any pair of arms $b_i$ and $b_j$, the probability of $b_i$ being preferred over $b_j$ is determined by $P(i \succ j) = \theta(i)/(\theta(i) + \theta(j))$. Setting the number of arms $K = 5$, we randomize the order of the arms to prevent selection bias, resulting in the following arm ordering: $b_5 \succ b_3 \succ b_2 \succ b_1 \succ b_4$. We use two instances: `Transitive-Easy` and `Transitive-Hard`, with their respective $\theta$ parameters given by:

- `Transitive-Easy` instance: $\theta(1) = 1$, $\theta(i) = 0.5 - (i-1)/2K$, $\forall i \in [2, K]$.

- `Transitive-Hard` instance: $\theta(i) = 1 - (i-1)/K, \ \forall i \in [K]$.

Note that the datasets generated in this way satisfy the Strong Stochastic Transitivity (SST) and Stochastic Triangle Inequality (STI) properties (Yue et al., 2012) (see Appendix B.3.1 for more details). The settings of the used BTL model also imply the existence of a Condorcet winner.

**Intransitive Case**: $CW \setminus (\text{SST} \cup \text{STI})$

In intransitive instances, the preference matrices are constructed to violate both the Strong Stochastic Transitivity (SST) and Stochastic Triangle Inequality (STI) properties. This design creates cyclic preferences among the non-winning arms while preserving the existence of a Condorcet winner. Setting $K = 5$, we still use the same shuffled arm ordering: $b_5 \succ b_3 \succ b_2 \succ b_1 \succ b_4$ for intransitive instances.

- `Intransitive-Easy` instance: The Condorcet winner $b_5$ has a strong preference over any other arm:
$$P(5 \succ j) = 0.8, \quad P(j \succ 5) = 0.2, \quad \forall j \in \{1, \ldots, 4\}.$$
Among the non-winning arms $b_1, \ldots, b_4$, cyclic preferences are introduced via:
$$P(i \succ j) = 0.8 - 0.2 \cdot ((j - i - 1) \bmod (K - 1)), \quad \forall i, j \in \{1, \ldots, 4\}, \ i \neq j.$$
This configuration ensures a clear dominance by $b_5$.

- `Intransitive-Hard` instance: The Condorcet winner's preference is weaker, with:
$$P(5 \succ j) = 0.6, \quad P(j \succ 5) = 0.4, \quad \forall j \in \{1, \ldots, 4\}.$$
This setting makes it more challenging to identify $b_5$ as the Condorcet winner.

Finally, in both instances, the symmetry condition is imposed for consistency:
$$P(j \succ i) = 1 - P(i \succ j), \quad \forall i, j \in \{1, \ldots, K\}, \ i \neq j.$$

Accordingly, as shown below, we create a cyclic pattern of preferences among the non-winning arms while maintaining the Condorcet winner's superiority.

*Intransitive-Easy Instance ($p_w = 0.8$)*

$$P = \begin{bmatrix} 0.0 & 0.8 & 0.6 & 0.4 & 0.2 \\ 0.2 & 0.0 & 0.8 & 0.6 & 0.2 \\ 0.4 & 0.2 & 0.0 & 0.8 & 0.2 \\ 0.6 & 0.4 & 0.2 & 0.0 & 0.2 \\ 0.8 & 0.8 & 0.8 & 0.8 & 0.0 \end{bmatrix}$$

*Intransitive-Hard Instance ($p_w = 0.6$)*

$$P = \begin{bmatrix} 0.0 & 0.8 & 0.6 & 0.4 & 0.4 \\ 0.2 & 0.0 & 0.8 & 0.6 & 0.4 \\ 0.4 & 0.2 & 0.0 & 0.8 & 0.4 \\ 0.6 & 0.4 & 0.2 & 0.0 & 0.4 \\ 0.6 & 0.6 & 0.6 & 0.6 & 0.0 \end{bmatrix}$$

## C.1.2 DESIGN OF PROMPTS

**Original Prompt**

System Prompt

**Problem Description:**
You are a Dueling Bandit algorithm with 5 arms. Each pair of arms (i, j) has an unknown probability P(i > j) of arm i winning against arm j in a pairwise duel. There exists a best arm (Condorcet Winner) among the 5 arms, which has a greater than 50% probability of winning against any other arm in a pairwise duel.

At each time step, I will provide you with a summary that shows the past dueling results and empirical probabilities. Then you must choose 2 arms to duel against each other. The reward you receive is the sum of the unknown probabilities of the two chosen arms beating the best arm. The maximum reward in each time step is achieved if the best arm duels against itself. Your goal is to maximize the cumulative sum of rewards of the chosen two arms over a given time horizon T=1000.

User Prompt

**Summarized History:**
So far, you have conducted {t} duels with the following results:
Arm 1 vs Arm 2: Arm 1 won 0 times, Arm 2 won 0 times, P(1 > 2) = 0.5, P(2 > 1) = 0.5
Arm 1 vs Arm 3: Arm 1 won 0 times, Arm 3 won 0 times, P(1 > 3) = 0.5, P(3 > 1) = 0.5
Arm 1 vs Arm 4: Arm 1 won 0 times, Arm 4 won 0 times, P(1 > 4) = 0.5, P(4 > 1) = 0.5
……
Arm 4 vs Arm 5: Arm 4 won 0 times, Arm 5 won 0 times, P(4 > 5) = 0.5, P(5 > 4) = 0.5

**Chain-of-Thought**
Let's think step by step to choose the next pair of arms.
Answer Format:
1. Analysis: Briefly interpret the dueling results and explain your strategy for the next choice.
2. Next Pair: (armA, armB).

Figure 7: Original prompt for LLM-Env interaction in dueling bandit setting with temperature = 0 (except O1-PREVIEW, which is in beta phase, its system prompt and user prompt are concatenated together with a fixed temperature = 1), including context $P$, summarized history $H_t$, and zero-shot chain-of-thought (CoT) reasoning instructions $R$ (see Section 2).

**Biased History**

Noisy User Prompt

**Summarized History:**
So far, you have conducted duels with the following results:
Arm 1 vs Arm 5: Arm 1 won +10 times,
Arm 2 vs Arm 5: Arm 2 won +10 times,
Arm 3 vs Arm 5: Arm 3 won +10 times,
Arm 4 vs Arm 5: Arm 4 won +10 times,
 ……
(All other text remains unchanged)

**Reversed Goal**

Adversarial System Prompt

**Problem Description:**
……
Your goal is to minimize the cumulative sum of rewards of the chosen two arms over a given time horizon T=2000.
……
(All other text remains unchanged)

Figure 8: Perturbed prompts used to evaluate the robustness of LEAD. The Biased History prompt (Left) injects an incorrect dueling history favoring non-optimal arms, while the Reversed Goal prompt (Right) adversarially modifies the objective from maximizing to minimizing reward. Both prompts maintain the zero-shot chain-of-thought (CoT) reasoning and temperature setting as before.

### C.1.3 EXEMPLARS OF GPT-4 TURBO AND O1-PREVIEW

We present exemplars using the original prompt (see Figure 7) to illustrate the decision-making process of both GPT-4 TURBO and O1-PREVIEW in the dueling bandits setting. These examples highlight how each model interprets the available data, reasons about the relative strengths of the arms, and selects the next pair of arms to duel. The exemplars show both successful convergence cases and the failures of each model.

### SYSTEMATIC BIASES

---

**Biased Understanding of GPT-4 TURBO and O1-PREVIEW**

**System Prompt:**

You are a Dueling Bandit algorithm with 5 arms. Each pair of arms (i, j) has an unknown probability $P(i > j)$ of arm $i$ winning against arm $j$ in a pairwise duel. There exists a best arm (Condorcet Winner) among the 5 arms, which has a greater than 50% probability of winning against any other arm in a pairwise duel. At each time step, I will provide you with a summary that shows the past dueling results and empirical probabilities. Then you must choose 2 arms to duel against each other. The reward you receive is the sum of the unknown probabilities of the two chosen arms beating the best arm. The maximum reward in each time step is achieved if the best arm duels against itself. Your goal is to maximize the cumulative sum of rewards of the chosen two arms over a given time horizon $T = 1000$.

**Question:** Can an arm duel against itself?

**Answer Exemplars:**
**Example 1**: In a traditional dueling bandit problem, an arm dueling with itself doesn't make practical sense because the outcome is trivially known—it would always win. The typical setup for dueling bandits is to choose two distinct arms to engage in a duel, which provides meaningful information about their relative strengths.
**Example 2**: In the context of a dueling bandit problem, an arm dueling with itself typically doesn't make sense because the purpose of a duel is to compare two different arms. Dueling an arm with itself wouldn't provide any new information about the relative strengths of the arms, as the outcome is trivially known: the arm would win against itself 100% of the time.
**Example 3**: In the context of a dueling bandit problem, an arm cannot duel with itself because the concept of a duel inherently involves comparing two different entities. The purpose of a duel is to establish which of the two arms is superior, or to collect comparative information. An arm dueling itself would not generate any valuable information or result, as it would trivially win against itself. Therefore, you need to choose two distinct arms for each duel in a dueling bandit problem.
**Example 4**: In the context of a dueling bandit problem, an arm cannot duel with itself because the concept of a duel inherently involves comparing two different entities. The purpose of a duel is to establish which of the two arms is superior, or to collect comparative information. An arm dueling itself would not generate any valuable information or result, as it would trivially win against itself. Therefore, you need to choose two distinct arms for each duel in a dueling bandit problem.

**Comment**: LLMs lack a fundamental understanding of the DB problem

---

SUCCESS CASES

---

**Successful Convergence Case of GPT-4 TURBO**

**1. Analysis:**
- Arm 5 appears to be the strongest contender, showing a high probability of winning against all other arms except Arm 2, where it still maintains a majority win rate.
- Arm 3 has a strong win rate against Arm 4 and a reasonable rate against Arm 2, but significantly loses to Arm 5.
- Arm 2 has a mixed performance, winning against Arm 1 and Arm 4 but losing to Arm 5 and Arm 3.
- Arm 1 and Arm 4 are the weakest, with Arm 1 losing to almost all arms and Arm 4 only winning against Arm 1.
- Given the goal to maximize the cumulative sum of rewards, focusing on duels involving Arm 5 seems most beneficial, as it has the highest probability of winning against other arms, suggesting it might be the Condorcet Winner.

**2. Next Pair:** (Arm 5, Arm 5).
- Dueling Arm 5 against itself maximizes the reward for this step, as it likely has the highest unknown probability of beating the best arm (itself in this case). This strategy ensures the highest possible reward in a single duel under the given conditions.

**Comment**: Successfully understand the problem and suggest a converged arm pair

---

**Successful Convergence Case of O1-PREVIEW**

**1. Analysis:**
Given the dueling results so far, we are tasked with maximizing the cumulative reward over the remaining time horizon $T = 2000$ steps. The reward at each time step is the sum of the unknown probabilities of the two chosen arms beating the best arm (the Condorcet winner). To maximize our reward, we need to identify the arm that is most likely the best based on the empirical data and choose it to duel against itself. This is because the maximum reward at each time step is achieved when the best arm duels against itself.
Analyzing the empirical probabilities:
- Arm 5 has beaten all other arms with a probability greater than 50%:
- $P(5 > 1) = 0.75$
- $P(5 > 2) = 0.75$
- $P(5 > 3) = 0.64$
- $P(5 > 4) = 1.00$
This suggests that Arm 5 is the most likely Condorcet winner based on the data.
Therefore, to maximize the cumulative reward, we should choose Arm 5 to duel against itself, as this maximizes the sum of probabilities $2 \times P(5 > \text{BestArm})$.

**2. Next Pair:**
(Arm 5, Arm 5)

**Comment**: Successfully understand the problem and suggest a converged arm pair

---

FAILURE CASES

---

**Convergence Instability Case of GPT-4 TURBO:**

**1. Analysis:**
- Arm 5 appears to be a strong contender as it has a high winning probability against most other arms, particularly against Arm 1, Arm 2, Arm 3, and Arm 4. Notably, it has a significant number of duels against Arm 3, which provides a robust estimate of its strength relative to Arm 3.
- Arm 3 also shows strength, especially against Arm 4 and Arm 2, but it has a lower winning probability against Arm 5.
- Arm 1 and Arm 4 seem to be weaker overall, with Arm 1 losing all duels against Arm 5 and Arm 4 having a low win rate against Arm 5.
- Arm 2 is somewhat in the middle, with a mixed record against the other arms.
Given the results, Arm 5 is likely the Condorcet Winner as it has a greater than 50% win rate against all other arms. To maximize the reward, it would be beneficial to continue testing Arm 5 against other strong contenders to confirm its dominance and further understand the dynamics between the other arms.

**2. Next Pair:**
- This pairing is chosen because while Arm 5 has shown strong performance against Arm 3, the large number of duels suggests that further testing could provide additional insights and confirm stability in the empirical probabilities. Additionally, dueling these two strong arms could potentially yield a high reward if Arm 5 continues to dominate.
Choose: (Arm 5, Arm 3)

**Comment**: Keeps comparing the same top arms without committing to a single winner

---

---

**Biased Understanding Case of O1-PREVIEW:**

**1. Analysis:**
From the given data, we can infer that Arm 5 is likely the best arm (Condorcet Winner) because it has beaten all other arms with probabilities greater than 50
- P(5 > 1) = 0.75
- P(5 > 2) = 0.75
- P(5 > 3) = 0.66
- P(5 > 4) = 1.00

Our goal is to choose two arms (armA and armB) such that the sum P(armA > best arm) + P(armB > best arm) is maximized.
From the empirical probabilities:
- P(3 > 5) = 0.34
- P(2 > 5) = 0.25
- P(1 > 5) = 0.25
- P(4 > 5) = 0.00
Arm 3 has the highest probability of beating Arm 5, followed by Arms 2 and 1. Arm 4 has a zero probability and should not be selected.
Between Arms 1 and 2, Arm 2 has participated in more duels against Arm 5, providing a more reliable estimate of its probability to beat the best arm. Additionally, Arm 2's performance against other arms suggests it may be slightly better than Arm 1.
Therefore, the best choice to maximize our expected reward is to select Arm 3 and Arm 2.

**2. Next Pair:**
(Arm 3, Arm 2)

**Comment**: Exhibits a biased understanding of the problem description: while successfully identifying the Condorcet Winner, it still selects suboptimal arms in duel.

---

## C.2 SUPPLEMENTARY EXPERIMENTS

### C.2.1 COMPARISONS WITH DIFFERENT METRICS

We present supplementary results to complement our case studies. (i) Figure 9 shows the strong and weak regret comparisons for the Hard instance. (ii) Figure 12 presents comparisons under different numbers of arms $K$, illustrating the impact of the Relative Decision Window. (iii) Figure 13 introduces the Best Arm Inclusion Ratio and the Converged Best Arm Ratio. (iv) Figure 14 examines the generalized variance of the strong and weak regret for both instances.

### C.2.2 DUEL SELECTION TRAJECTORY

We visualize the duel selection trajectory in representative experiments to better understand the behavior of LLM agents and baseline algorithms.

**Duel Selection Trajectory Explanation:** The reshuffled arm order is $b_5 \succ b_3 \succ b_2 \succ b_1 \succ b_4$, with arm indices from bottom to top: 5, 4, 3, 2, 1. Each filled black cell represents a selected arm at that time step. For instance, black lines in arms 5 and 3 indicate the selection of the duel between (arm 5, arm 3) at that particular time step.

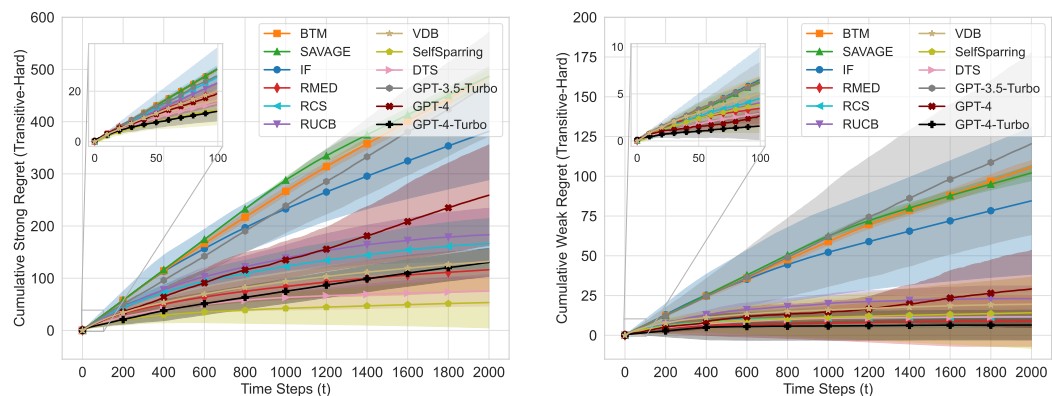

Figure 9: Comparisons between LLM agents and various classic DB algorithms. **Left** and **Right**: strong and weak regret for the `Transitive-Hard` instance. Results for the `Transitive-Easy` instance is presented in Figure 2. We evaluate only three LLMs on the `Transitive-Hard` instance due to our research goals and high API costs: (i) The results for the `Transitive-Hard` instance are qualitatively similar to those for the `Transitive-Easy` instance; (ii) Obviously, the `Transitive-Easy` instance offers higher distinguishability, allowing us to observe convergence and regret differences within a feasible number of steps.

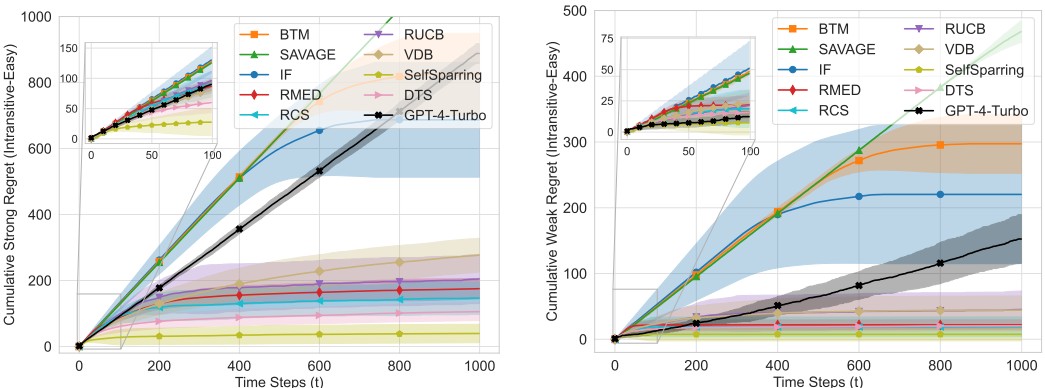

Figure 10: Comparisons between GPT-4 TURBO and various classic DB algorithms. **Left** and **Right**: strong and weak regret for the `Intransitive-Easy` instance. Results for the `Intransitive-Hard` instance is presented in Figure 11. We evaluate only our top-performing LLM on the `Intransitive-Easy` and `Intransitive-Hard` instance to examine the scalability limitation.

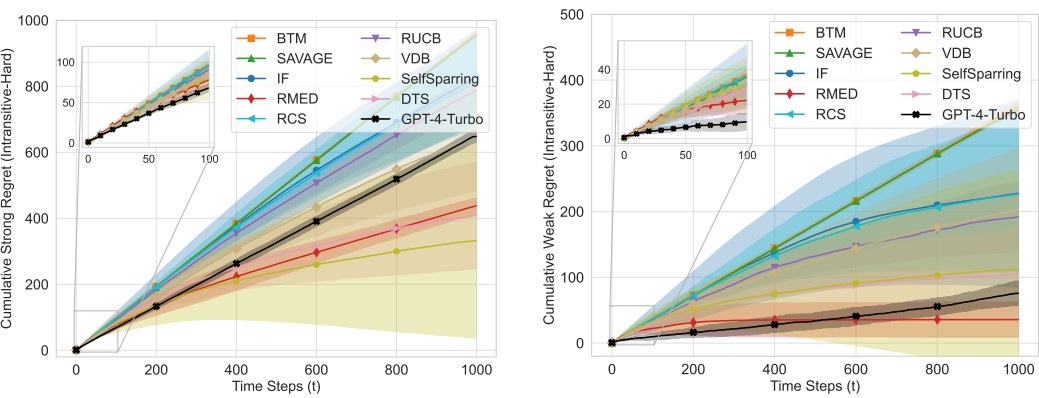

Figure 11: Comparisons between gptc and various classic DB algorithms. **Left** and **Right**: strong and weak regret for the `Intransitive-Hard` instance.

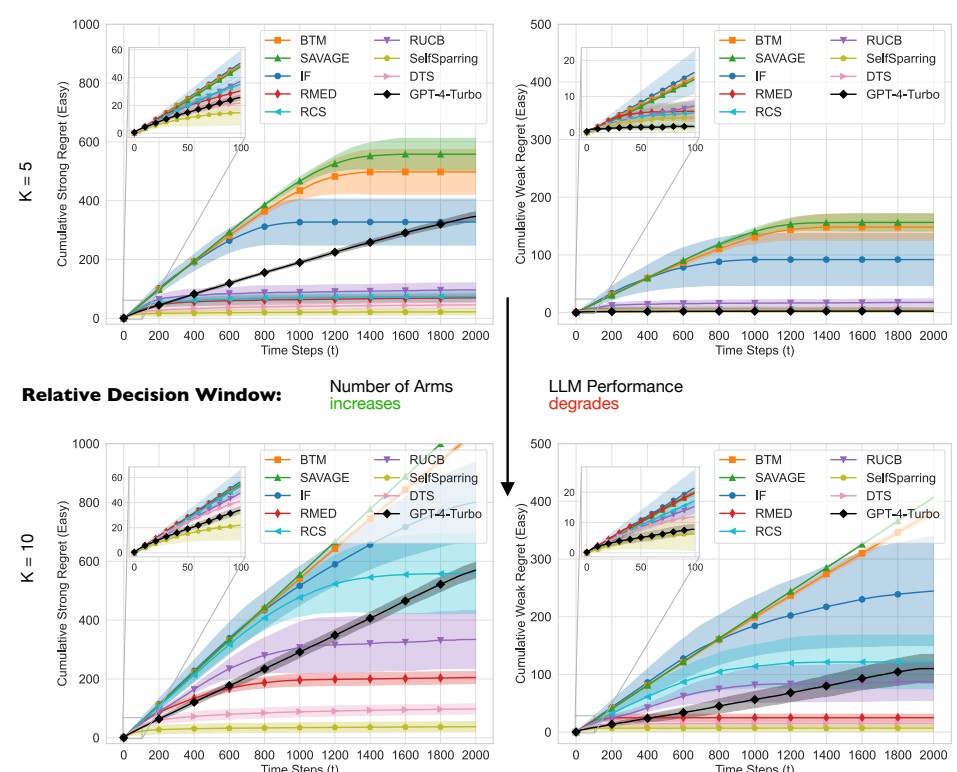

Figure 12: Cumulative strong and weak regret comparisons between LLM agents and classic dueling bandit algorithms on `Transitive-Easy` instance under different numbers of arms K. **Top Left** and **Top Right**: K=5, where GPT-4-Turbo significantly outperforms other methods on weak regret. **Bottom Left** and **Bottom Right**: K=10, where the performance of GPT-4-Turbo degrades as the number of arms increases.

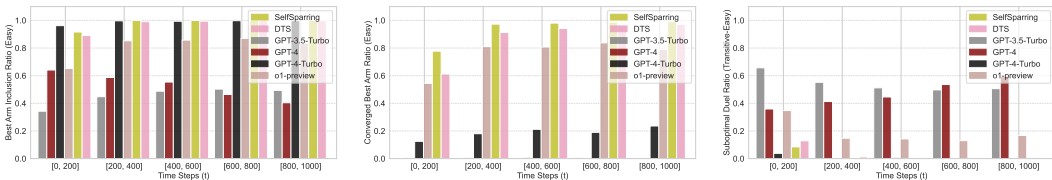

Figure 13: Four LLMs (GPT-3.5 TURBO, GPT-4, GPT-4 TURBO, O1-PREVIEW) and two state-of-the-art baselines (SELF-SPARRING and DTS) are compared against each other on the `Transitive-Easy` instance over different time intervals. **Left**: the Best Arm Inclusion Ratio represents the fraction of duels that include the best arm (Condorcet winner). **Middle**: the Converged Best Arm Ratio represents the proportion of duels where the best arm duels against itself for exploitation. **Right**: the Suboptimal Duel Ratio represents the proportion of duels where both arms selected in duel are suboptimal arms. We observed that while O1-PREVIEW can transit from exploration to exploitation (high Converged Best Arm Ratio), it selects more optimal arms (high Suboptimal Duel Ratio) due to the reinforced biased understanding as discussed in Section 3.2.

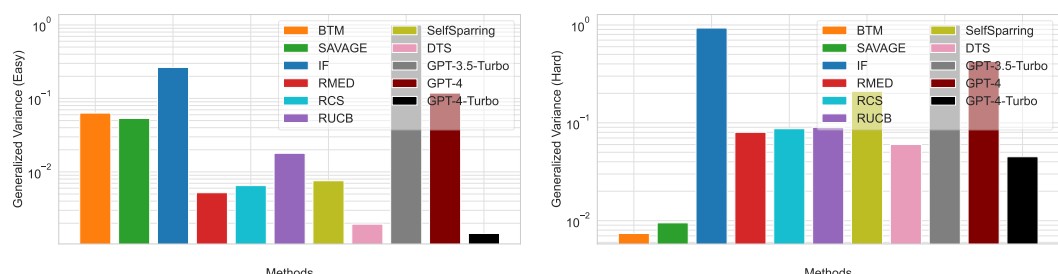

Figure 14: Comparison of the generalized variance of strong and weak regret between three LLMs and baseline algorithms on the `Transitive-Easy` (Left) and `Transitive-Hard` (Right) instances. In the `Easy` instance, GPT-4 TURBO exhibits the lowest average generalized variance. For the `Transitive-Hard` instance, GPT-4 TURBO maintains a variance level comparable to state-of-the-art baseline algorithms (except BTM and SAVAGE, which are in an early exploration stage).

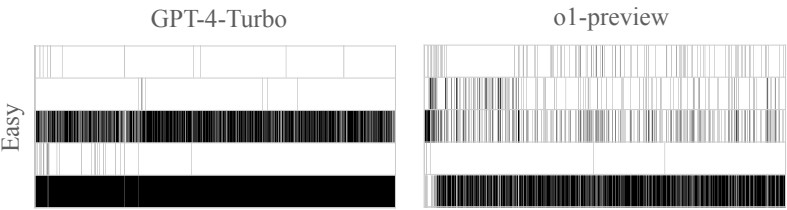

Figure 15: Comparison of duel selection trajectories between GPT-4 TURBO (Left) and O1-PREVIEW (Right) on the `Transitive-Easy` instance. GPT-4 TURBO achieves low weak regret by consistently selecting the best arm, though it struggles to converge to a single best arm. In contrast, O1-PREVIEW shows better convergence behavior, but its weak regret performance is worse than GPT-4 TURBO due to incomplete or biased understanding, as illustrated by the O1-PREVIEW exemplar in Appendix C.1.3.

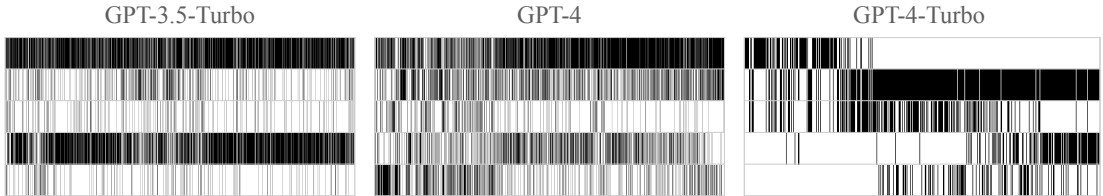

Figure 16: Local optima trajectories of GPT-3.5 TURBO (Left), GPT-4 (Middle), and GPT-4 TURBO (Right, with noisy prompt) on the `Transitive-Hard` instance. Less capable LLMs, such as GPT-3.5 TURBO and GPT-4, could get stuck comparing suboptimal arms on hard preference structure. Even for GPT-4 TURBO, noisy prompts with biased history (see Figure 8) can lead it to be trapped in bad tournaments.

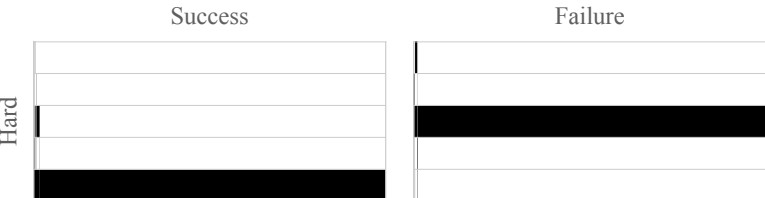

Figure 17: Comparison of success (Left) and failure (Right) cases for the Convergence-Triggered GPT-4 TURBO intervention strategy discussed in Section 4.1. While it works for most cases due to GPT-4 TURBO's strong capability (Left), sometimes this naive intervention can reinforce suboptimal choices (Right) on the `Transitive-Hard` instance.

