# OpenReview forum: "Beyond Numeric Awards: In-Context Dueling Bandits with LLM Agents"
_ICLR.cc/2025/Conference — Submitted to ICLR 2025_

### Official Review · Reviewer_FhH8 · 2024-10-29

**Soundness:** 3
**Presentation:** 3
**Contribution:** 3
**Rating:** 5
**Confidence:** 3

**Summary:**

This submission studies the abilities of large language models (LLMs) in dueling bandit settings. In dueling bandits, a decision-maker (in this case, an LLM) chooses a pair of arms at each time-step, but only observes the binary outcome of a noisy comparison between the two selected arms. This submission asks whether LLMs are effective in-context agents for solving dueling bandit problems.

In dueling bandits, the learner selects 2 of K arms for a noisy comparison at each round, i.e. a duel. The probability an arm wins a duel is proportional to how “distinguishable” the two arms are from one another. The dueling bandit problem is given to the LLM via a natural language description, with an externally-summarized history, and the LLM is prompted to use zero-shot chain-of-thought reasoning. In dueling bandits, the goal is to maximize the cumulative reward over the time horizon, where the reward is the sum of the (unknown) probabilities that the two chosen arms are the best arm. The performance of dueling bandit algorithms are measured by strong and weak regret. Strong regret compares the cumulative performance of the best arm against both arms chosen by the algorithm, and weak regret compares the best arm against the better of the two selected arms.

The authors empirically compare 5 LLMs against 8 baseline dueling bandit algorithms in 2 stochastic environments. They find that GPT-3.5 Turbo and GPT-4 fail to solve the dueling bandits problem, but GPT-4 Turbo outperforms state-of-the-art dueling bandit algorithms in terms of weak regret. Across all LLMs, they find that performance degrades as the number of arms increases. They capture this via the notion of a relative distance window, which is the maximum number of arms that an LLM agent can effectively handle in a dueling bandit problem while maintaining state-of-the-art performance.

Somewhat surprisingly, the authors find that GPT-4 turbo achieves lowest weak regret among all LLMs & SOTA baselines, but it struggles to converge to a single best arm. In contrast, O1-preview shows better strong regret performance than GPT-4 turbo, but its weak regret performance is significantly worse. The authors conclude that GPT-4 turbo can serve as an effective decision-maker in the short term, by quickly identifying and exploiting the best arm with low variance across different instances. However, its long-term performance is hindered by over-estimation bias and lack of convergence.

Motivated by these findings, the authors propose an algorithm (LEAD) for using LLMs to solve dueling bandit problems. The algorithm has two phases. In phase 1, the algorithm starts with a set of candidate options. Given two options suggested by the language model, the algorithm identifies a winner between them. This winning option is then compared with each remaining option in the candidate set until it either loses or all options have been matched. If the winning option remains undefeated, a "TrustLLM" flag is set to True; if it is defeated, the flag is set to False. In Phase 2, if the winning option from Phase 1 is defeated, the algorithm performs one round of a secondary selection process, using an estimated preference matrix to choose an "incumbent" option. After Phase 2, the algorithm returns to Phase 1, repeating the process until only the best option remains in the candidate set.

Under an assumption that the worst-case behavior of the LLM is equivalent to a randomizer that selects action pairs uniformly-at-random, the authors show that there exists an attacker strategy such that any standalone LLM agent will suffer high expected regret. The authors also prove theoretical guarantees for LEAD under an assumption on the abilities of the LLM.

Empirically, the authors find that LEAD exhibits competitive performance with baselines in terms of strong regret, and LEAD has superior performance in terms of weak regret. They also show that LEAD can overcome a biased history, whereas a standalone LLM cannot.

**Strengths:**

Studying the abilities of LLM agents in in-context reinforcement learning problems is an exciting new direction. While the authors are not the first to do this, they are the first to consider the ability of LLMs in dueling bandits problems. The authors provide a fairly comprehensive set of empirical results, both in benchmarking the performance of LLM agents and proposing a mitigation (LEAD) to improve their performance. The submission is also well-written for the most part.

**Weaknesses:**

I found the theoretical results to be somewhat unsatisfying. For one, while the lower bound is correct as-stated, it seems to not capture the spirit of the problem, as (presumably) the reason to use an LLM to solve a decision-making task is that the decision-maker belives the LLM has learned some useful knowledge about said problem somewhere during its training. As such, if the problem instance resembles something adversarial, it would probably be better to deploy an off-the-shelf algorithm for dueling bandits.

I also had a hard time parsing Theorem 4.2. In particular, the regret bounds only kick in when the two arms the LLM recommends always contains the best arm. This seems overly restrictive, and would probably not hold in practice. In fact, is this property even known to be satisfied for any off-the-shelf dueling bandit algorithms?

**Questions:**

Is my understanding of the assumption in Theorem 4.2 correct?

---

> ### Author Response · Authors · 2024-11-28
> **Rebuttal by Authors (Part I)**
>
> Thank you for your positive feedback on the originality and significance of our work. Please see our responses to the comments raised in your review below:
>
> 1. >  the reason to use an LLM to solve a decision-making task is that the decision-maker belives the LLM has learned some useful knowledge about said problem somewhere during its training.
>
> Thank you so much for your feedback. We have noticed that the presentation in some parts of the previous manuscript was indeed unclear and caused misunderstandings. We have revised some of the key points of the motivation/introduction/conclusion to achieve a clearer positioning. According to the updated manuscript, **we want to emphasize that we are exploring the capability boundaries of the in-context dueling bandit with general-purpose LLM agents**, contributing to the problem of In-Context Reinforcement Learning (ICRL) [1, 2, 3, 4].
>
> While task-specific training of large sequence models can yield promising ICRL results [1, 2], it is often impractical due to the substantial computational resources required. **Similar to the settings in [3, 4, 5, 6, 7], we evaluate the emergent zero-shot abilities of ICRL in general-purpose LLMs under the dueling bandit problem, without re-training or fine-tuning.** This work focuses on understanding the fundamental capabilities and limitations of in-context dueling bandit with LLMs, as this knowledge will guide the development of more robust and versatile foundation agents in the future [8]. We have revised our manuscript to emphasize our positioning and motivation.
>
> 2. > As such, if the problem instance resembles something adversarial, it would probably be better to deploy an off-the-shelf algorithm for dueling bandits.
>
> You are right that if the prompt resembles something adversarial, off-the-shelf DB algorithms may indeed perform better. However, the ability to effectively handle unknown adversarial scenarios in in-context decision-making is a practical requirement for deploying these agents in real-world applications [8], **especially under serious scenarios where unknown adversarial attacks are guaranteed to exist**. **A safe and robust AI system under serious scenarios has to be provably safe under worst-case conditions** [9, 10]. This highlights the importance of constructing robust in-context decision-making agents that can adaptively handle such situations. Our work contributes to this by offering insights into how an agentic system, despite the black-box nature of LLM, can achieve effective and robust performance through the interplay between rule-based experts and in-context LLM agents.
>
> 3. > The regret bounds only kick in when the two arms the LLM recommends always contains the best arm. This seems overly restrictive, and would probably not hold in practice. In fact, is this property even known to be satisfied for any off-the-shelf dueling bandit algorithms?
>
> Yes, this property is satisfied for all the explore-then-exploit Dueling Bandit algorithms. To clarify, Dueling Bandit algorithms can be broadly classified into two categories as Explore-Then-Exploit methods and Ongoing Regret Minimization methods according to [11]. Explore-Then-Exploit methods focus on identifying the best arm with high confidence before exploiting it, such as IF2 and BTM, etc. In contrast, Ongoing Regret Minimization methods explicitly target the objective of minimizing cumulative regret, including RUCB and Self-Sparring, etc. For Explore-Then-Exploit methods,  firstly the algorithm aims to identify the best arm $\hat{b}$ during the exploration phase; and after identifying $\hat{b}$, the algorithm enters the exploitation phase, repeatedly selecting ($\hat{b}$, $\hat{b}$) (i.e., the best arm duel against itself) for the remaining rounds [12]. If the identification of $\hat{b}$ during the explore phase is accurate, the exploit phase will theoretically incur no additional regret. In summary, our paper introduces an agentic flow framework, LLM with Enhanced Algorithmic Dueling (LEAD) that integrates off-the-shelf Explore-then-Exploit DB algorithms with LLM agents. This framework enables the fine-grained adaptive interplay between rule-based expert systems and incontext LLM agents, enhancing their ability to handle DB problems via algorithmic interventions as suggested in [3, 4].
>
> Thank you so much for your insightful feedback. In considering these responses to your feedback, we hope you will consider increasing your score for our paper.

---

> ### Author Response · Authors · 2024-11-28
> **Rebuttal by Authors (Part II)**
>
> [1] Laskin, Michael, et al. "In-context reinforcement learning with algorithm distillation." arXiv preprint arXiv:2210.14215 (2022).
>
> [2] Lee, Jonathan, et al. "Supervised pretraining can learn in-context reinforcement learning." Advances in Neural Information Processing Systems 36 (2024).
>
> [3] Krishnamurthy, Akshay, et al. "Can large language models explore in-context?." arXiv preprint arXiv:2403.15371 (2024).
>
> [4] Nie, Allen, et al. "EVOLvE: Evaluating and Optimizing LLMs For Exploration." arXiv preprint arXiv:2410.06238 (2024).
>
> [5] Mirchandani, Suvir, et al. "Large language models as general pattern machines." arXiv preprint arXiv:2307.04721 (2023).
>
> [6] Chen, Dingyang, Qi Zhang, and Yinglun Zhu. "Efficient Sequential Decision Making with Large Language Models." arXiv preprint arXiv:2406.12125 (2024).
>
> [7] Benechehab, Abdelhakim, et al. "Zero-shot Model-based Reinforcement Learning using Large Language Models." arXiv preprint arXiv:2410.11711 (2024).
>
> [8] Liu, Xiaoqian, et al. "Position: Foundation Agents as the Paradigm Shift for Decision Making." arXiv preprint arXiv:2405.17009 (2024).
>
> [9] Dalrymple, David, et al. "Towards Guaranteed Safe AI: A Framework for Ensuring Robust and Reliable AI Systems." arXiv preprint arXiv:2405.06624 (2024).
>
> [10] Tegmark, Max, and Steve Omohundro. "Provably safe systems: the only path to controllable AGI." arXiv preprint arXiv:2309.01933 (2023).
>
> [11] Zoghi, Masrour, et al. "Relative confidence sampling for efficient on-line ranker evaluation." Proceedings of the 7th ACM international conference on Web search and data mining. 2014.
>
> [12] Yue, Yisong, et al. "The k-armed dueling bandits problem." Journal of Computer and System Sciences 78.5 (2012): 1538-1556.

---

> ### Author Response · Authors · 2024-12-03
> **Gentle Reminder for Your Valuable Feedback**
>
> I hope this finds you well. This is a gentle reminder that there is less than one day remaining before the discussion period concludes. We would greatly appreciate any further comments or feedback you might have on our revisions to ensure all concerns are thoroughly addressed.
>
> Thanks for your time and considerations.
>
> Best regards,
>
> The Authors

---

### Official Review · Reviewer_dSFz · 2024-11-02

**Soundness:** 3
**Presentation:** 2
**Contribution:** 2
**Rating:** 3
**Confidence:** 3

**Summary:**

The authors explore the use of LLMs in the dueling bandits problem. They find that GPT-4 Turbo is the strongest LLMs in their experiments. They create a dueling bandit algorithm that can take LLM advice initially but then switches to an algorithm with guarantees if the advice goes bad. By tuning hyperparameters, they can get their LLM advice guided algorithm to perform on par with the strongest competing (LLM-less methods).

**Strengths:**

- They are able to achieve strong empirical performance with an LLM-advised algorithm that still maintains worst-case theoretic guarantees. This could be especially valuable in the setting where there is text or image context that is provided for each arm that can be exploited by an LLM or VLM. For example, in an information retrieval setting, each arm could come with text describing what it is returned, and the context could be the search query.

**Weaknesses:**

- Right now, the motivation of the paper is not very clear. There seems to be no reason to want to use an LLM-guided dueling bandit algorithm unless there is text or image context for the arms. None of the experiments have this, and the authors do not discuss what this could be.

As is, there is no performance benefit to the LLM-guided version—it is slower, more expensive, and does random stuff sometimes. Yes, it can perform comparably sometimes, but requires hyperparameter tuning to do so.
- The presentation is bad—flow is awkward, lots of key details are deep in the appendix (for example, the MatchArms procedure). Much of the main body text is not very impactful. There are many citations that should be parenthetical.
- From an LLMology perspective, the central mystery is why o1 preview is worse than GPT 4 here. This problem should be easily amenable to o1 style synthetic data, so we can probably conclude that it wasn't included in the o1 training data, but still, the extended CoT should be helpful. The LLMologists would probably appreciate some analysis of when o1 preview fails.

**Questions:**

1. Why is it important to develop an LLM-guided algorithm for dueling bandits?

---

> ### Author Response · Authors · 2024-11-28
> **Rebuttal by Authors (Part I)**
>
> We thank reviewer dSFz for pointing out several points of inclarity in the existing manuscript. We now address each of the points in your review in turn:
>
> 1. > Right now, the motivation of the paper is not very clear. There seems to be no reason to want to use an LLM-guided dueling bandit algorithm unless there is text or image context for the arms. None of the experiments have this, and the authors do not discuss what this could be.
>
> Thank you so much for your feedback. We have noticed that the presentation in some parts of the previous manuscript was indeed unclear and caused misunderstandings. We have revised some of the key points of the motivation/introduction/conclusion to achieve a clearer positioning. According to the updated manuscript, we want to emphasize that **we are not developing a new LLM-augmented dueling bandits algorithm but are instead exploring the capability boundaries of the in-context dueling bandit with general-purpose LLM agents**, contributing to the problem of In-Context Reinforcement Learning (ICRL) [1, 2, 3, 4]. For (contextual) MAB problems, [4] found that an optimality gap persists between LLMs and MAB algorithms even with inference-time algorithmic guidance, which is consistent with our results on strong regret. Our proposed agentic framework via a best-of-both-worlds design bridges this gap through fine-grained adaptive interplay. The framework sheds light on how language-based reasoning can inspire provably robust frameworks [5, 6] that translate words into actions, paving the way for more trustworthy AI systems through the interplay between rule-based experts and in-context LLM agents.
>
> As for the extra text/image context you mentioned, our scope is to evaluate LLMs as in-context decision-makers for standard context-free dueling bandits (DB) with a Condorcet Winner, offering the first systematic insights into their strengths and limitations. Similar to the settings in [3, 4, 10], we aim to understand the strategic behavior of general-purpose LLMs within such a context-free setting. However, we acknowledge that when the prompt contains extra textual/image context that can infer the relative preferences between arms, the $T_{LLM}$ will decrease, further enhancing the best-case performance. And we have mentioned in the manuscript that we consider it an important direction for future work within the Contextual Dueling Bandit framework [11].
>
> 2. > there is no performance benefit to the LLM-guided version—it is slower, more expensive, and does random stuff sometimes. Yes, it can perform comparably sometimes, but requires hyperparameter tuning to do so.
>
> Your insight is completely right in some sense. However, I want to cite the paragraph below in [7] to answer this question: "**While difficult to deploy today for real systems due to latency, context size limitations, and compute costs, the approach of using LLMs to drive low-level control may provide an exciting glimpse into how the patterns among words could be transferred to actions.**"
>
> We would also like to borrow praise of our motivation from other reviewers:
>
> - Reviewer FhH8: Studying the abilities of LLM agents in in-context reinforcement learning problems is an exciting new direction.
>
> - Reviewer G8sE: I believe there is currently quite some interest in evaluations of in-context decision-making capabilities of LLMs w.r.t. exploratory and interactive learning behavior
>
> The exploration of foundation agents' in-context decision making ability, even with their current limitations, is a valuable and necessary step towards AGI [8]. There has been a large and fast-growing body of literature on LLM agents for decision-making, where LLMs are employed as online decision-makers in various real-world applications. This includes tasks such as (contextual) MAB [3, 4], Game Theory [9], Model-Based RL [10], etc. Our work contributes to this line of research by focusing on Dueling Bandits. The focus, for now, shouldn't be solely on beating traditional algorithms theoretically/empirically/economically. Rather, it should be on understanding the fundamental capabilities and limitations of in-context decision-making with LLMs, as this knowledge will guide the development of more robust and versatile foundation agents in the future [8].

---

> ### Author Response · Authors · 2024-11-28
> **Rebuttal by Authors (Part II)**
>
> 3. > The presentation is bad—flow is awkward, lots of key details are deep in the appendix (for example, the MatchArms procedure). Much of the main body text is not very impactful. There are many citations that should be parenthetical.
>
> Thank you very much for your valuable feedback. We have almost completely revised the presentation flow, moved the positions of the figures, and resummarized the key findings to clearly convey our results. All changes can be seen in the new manuscript.
>
> 4. > From an LLMology perspective, the central mystery is why o1 preview is worse than GPT 4 here. This problem should be easily amenable to o1 style synthetic data, so we can probably conclude that it wasn't included in the o1 training data, but still, the extended CoT should be helpful. The LLMologists would probably appreciate some analysis of when o1 preview fails.
>
> Thank you for pointing this out. In the System Biases section on Page 5, we have added an explanation for why o1 performs worse than 4turbo, included failure exemplar (Appendix C.1.3), and provided a comparison of their related ratios (Figure 13). GPT-4 Turbo and o1-preview exhibit systematic biases regarding the DB problem, likely due to a lack of exposure to similar tasks during pretraining. Specifically, they incorrectly assume that an arm cannot duel with itself (the convergence case), even when explicitly prompted to do so (see examples in Appendix C.1.3). This misunderstanding makes the DB problem as an out-of-distribution (OOD) task for LLMs, and in-context instructions fail to fully override this internal bias. Consequently, LLM agents cannot completely align with problem descriptions due to the inherent limitations of in-context learning, which cannot really generalize to OOD tasks [12]. Figure 13 supports these observations: o1-preview demonstrates better reasoning capabilities by transitioning from exploration to exploitation effectively and achieving lower strong regret than GPT-4 Turbo and o1-preview. However, its CoT mechanism reinforces its internal biased understanding of DB, resulting in poorer
>
> Again, **thank you so much for your insightful feedback to help us identify the lack of clarity in the original manuscript and made corresponding revisions**. Based on these additional clarifications, we hope you will consider increasing your score in support of this work. Otherwise, could you let us know what additional changes or improvements are necessary to make this work ready for publication?
>
> [1] Laskin, Michael, et al. "In-context reinforcement learning with algorithm distillation." arXiv preprint arXiv:2210.14215 (2022).
>
> [2] Lee, Jonathan, et al. "Supervised pretraining can learn in-context reinforcement learning." Advances in Neural Information Processing Systems 36 (2024).
>
> [3] Krishnamurthy, Akshay, et al. "Can large language models explore in-context?." arXiv preprint arXiv:2403.15371 (2024).
>
> [4] Nie, Allen, et al. "EVOLvE: Evaluating and Optimizing LLMs For Exploration." arXiv preprint arXiv:2410.06238 (2024).
>
> [5] Dalrymple, David, et al. "Towards Guaranteed Safe AI: A Framework for Ensuring Robust and Reliable AI Systems." arXiv preprint arXiv:2405.06624 (2024).
>
> [6] Tegmark, Max, and Steve Omohundro. "Provably safe systems: the only path to controllable AGI." arXiv preprint arXiv:2309.01933 (2023).
>
> [7] Mirchandani, Suvir, et al. "Large language models as general pattern machines." arXiv preprint arXiv:2307.04721 (2023).
>
> [8] Liu, Xiaoqian, et al. "Position: Foundation Agents as the Paradigm Shift for Decision Making." arXiv preprint arXiv:2405.17009 (2024).
>
> [9] Park, Chanwoo, et al. "Do llm agents have regret? a case study in online learning and games." arXiv preprint arXiv:2403.16843 (2024).
>
> [10] Benechehab, Abdelhakim, et al. "Zero-shot Model-based Reinforcement Learning using Large Language Models." arXiv preprint arXiv:2410.11711 (2024).
>
> [11] Dudík, Miroslav, et al. "Contextual dueling bandits." Conference on Learning Theory. PMLR, 2015.
>
> [12] Wang, Qixun, et al. "Can In-context Learning Really Generalize to Out-of-distribution Tasks?." arXiv preprint arXiv:2410.09695 (2024).

---

> ### Author Response · Authors · 2024-12-03
> **Gentle Reminder for Your Valuable Feedback**
>
> I hope this finds you well. This is a gentle reminder that there is less than one day remaining before the discussion period concludes. We would greatly appreciate any further comments or feedback you might have on our revisions to ensure all concerns are thoroughly addressed.
>
> Thanks for your time and considerations.
>
> Best regards,
>
> The Authors

---

### Official Review · Reviewer_YNcf · 2024-11-02

**Soundness:** 3
**Presentation:** 2
**Contribution:** 2
**Rating:** 6
**Confidence:** 4

**Summary:**

This paper investigates the capabilities of Large Language Models (LLMs) as decision-makers in the dueling bandits setting, focusing on preference feedback instead of numerical rewards. The authors evaluate models like GPT-4 TURBO and compare their performance against classic DB algorithms. While GPT-4 TURBO demonstrates strong short-term decision-making by identifying the Condorcet winner with low weak regret, it struggles with long-term convergence and prompt sensitivity. To address these limitations, the authors introduce LEAD (LLM-Enhanced Adaptive Dueling), a hybrid algorithm combining LLM capabilities with the theoretical guarantees of traditional DB algorithms. LEAD shows improved robustness and efficacy, even under noisy or adversarial prompts.

**Strengths:**

The paper uniquely applies LLMs to a DB context, demonstrating their strengths and vulnerabilities.

Comprehensive experiments show how LLMs compare with state-of-the-art DB algorithms and the advantages of the LEAD framework.

LEAD is designed with strong theoretical underpinnings that ensure bounded regret, which is essential for real-world applications.

**Weaknesses:**

The LEAD algorithm essentially combines LLM capabilities with the IF2 algorithm, which might be perceived as a straightforward application rather than a fundamentally new algorithm.

**Questions:**

1. The paper mentions theoretical guarantees for LEAD. Are these guarantees entirely new, or do they largely extend those of IF2? If novel, could you explain the key differences and how these guarantees are specific to the hybrid nature of LEAD?

2. How does the LEAD algorithm scale when the number of arms or the problem complexity increases? Are there any significant computational trade-offs when integrating LLMs with IF2, particularly in larger decision-making environments?

3. The paper focuses on dueling bandit problems. Can the approach be generalized to other types of decision-making or preference-based learning problems? If so, are there adjustments needed to apply LEAD outside the DB context?

---

> ### Author Response · Authors · 2024-11-28
> **Rebuttal by Authors (Part I)**
>
> Thank you for your positive feedback on the originality and significance of our work. Please see our responses to the comments raised in your review below:
>
> 1. > The LEAD algorithm essentially combines LLM capabilities with the IF2 algorithm, which might be perceived as a straightforward application rather than a fundamentally new algorithm.
>
> Thank you so much for your feedback. We have noticed that the presentation in some parts of the previous manuscript was unclear and caused misunderstandings. We have revised some of the key points of the motivation/introduction/conclusion to achieve a clearer positioning. According to the updated manuscript, we want to emphasize that **we are not developing a new LLM-augmented dueling bandits algorithm but are instead exploring the capability boundaries of the in-context dueling bandit with general-purpose LLM agents**, contributing to the problem of In-Context Reinforcement Learning (ICRL) [1, 2, 3, 4]. For (contextual) MAB problems, [4] found that an optimality gap persists between LLMs and MAB algorithms even with inference-time algorithmic guidance, which is consistent with our results on strong regret. Our proposed agentic framework via a best-of-both-worlds design bridges this gap through fine-grained adaptive interplay. The framework sheds light on how language-based reasoning can inspire provably robust frameworks [6, 7] that translate words into actions, paving the way for more trustworthy AI systems through the interplay between rule-based experts and in-context LLM agents.
>
> 2. > The paper mentions theoretical guarantees for LEAD. Are these guarantees entirely new, or do they largely extend those of IF2? If novel, could you explain the key differences and how these guarantees are specific to the hybrid nature of LEAD?
>
> The framework's theoretical guarantees are inherited from IF2 (or can be other explore-then-exploit algorithms) **with non-trivial algorithmic interventions in the agentic framework design, as suggested in [3, 4, 6, 7]**. LEAD is explicitly designed to leverage the strengths of both standalone LLMs and robust DB algorithms. It benefits from the exploratory behavior of the LLM within the Relative Decision Boundary (RDB). The LLM's zero-shot reasoning capabilities allow it to prioritize promising arms during initial exploration, which reduces the exploration burden on the underlying DB algorithm.
>
> 3. > How does the LEAD algorithm scale when the number of arms or the problem complexity increases? Are there any significant computational trade-offs when integrating LLMs with IF2, particularly in larger decision-making environments?
>
> The revised form of the definition of Relative Decision Boundary (RDB) refers to the definition of the CoT Reasoning Boundary (RB) definition in this NIPS oral paper [5]. We want to quantify the generalization failure (LLMs' performance degrades when introducing intransitive preference structures or a large number of arms) of in-context LLM agents. RDB provides an empirically grounded definition to capture this phenomenon under the influence of relevant factors. However, as the first step to evaluate in-context DB, quantifying and experimenting with each factor's specific impact on RDB is out of scope for our work (e.g., [5] dedicated an entire paper to designing experiments to draw related conclusions of CoT). We consider it an important future work direction.
>
> [1] Laskin, Michael, et al. "In-context reinforcement learning with algorithm distillation." arXiv preprint arXiv:2210.14215 (2022).
>
> [2] Lee, Jonathan, et al. "Supervised pretraining can learn in-context reinforcement learning." Advances in Neural Information Processing Systems 36 (2024).
>
> [3] Krishnamurthy, Akshay, et al. "Can large language models explore in-context?." arXiv preprint arXiv:2403.15371 (2024).
>
> [4] Nie, Allen, et al. "EVOLvE: Evaluating and Optimizing LLMs For Exploration." arXiv preprint arXiv:2410.06238 (2024).
>
> [5] Chen, Qiguang, et al. "Unlocking the Capabilities of Thought: A Reasoning Boundary Framework to Quantify and Optimize Chain-of-Thought." arXiv preprint arXiv:2410.05695 (2024).
>
> [6] Dalrymple, David, et al. "Towards Guaranteed Safe AI: A Framework for Ensuring Robust and Reliable AI Systems." arXiv preprint arXiv:2405.06624 (2024).
>
> [7] Tegmark, Max, and Steve Omohundro. "Provably safe systems: the only path to controllable AGI." arXiv preprint arXiv:2309.01933 (2023).

---

> ### Author Response · Authors · 2024-11-28
> **Rebuttal by Authors (Part II)**
>
> 4. > The paper focuses on dueling bandit problems. Can the approach be generalized to other types of decision-making or preference-based learning problems? If so, are there adjustments needed to apply LEAD outside the DB context?
>
> We added the clarification that DB is a stateless preference-based reinforcement learning setting [3, 4] (which is a preference-based version of MAB, extended from [1, 2]) by querying for preference feedback. However, stateless RL and state-based RL are two different settings and cannot be confused with each other. Specifically, our scope is the part of stateless preference-based RL but not the state-based ones.
>
> We hope that these clarifications effectively communicate the positioning of our work. Thank you once again for your useful comments. In considering these clarifications for your feedback, would you consider increasing your score for our paper? If not, could you let us know any additional changes you would like to see in order for this work to be accepted?
>
> [1] Krishnamurthy, Akshay, et al. "Can large language models explore in-context?." arXiv preprint arXiv:2403.15371 (2024).
>
> [2] Nie, Allen, et al. "EVOLvE: Evaluating and Optimizing LLMs For Exploration." arXiv preprint arXiv:2410.06238 (2024).
>
> [3] Wirth, Christian, et al. "A survey of preference-based reinforcement learning methods." Journal of Machine Learning Research 18.136 (2017): 1-46.
>
> [4] Saha, Aadirupa, Aldo Pacchiano, and Jonathan Lee. "Dueling rl: Reinforcement learning with trajectory preferences." International Conference on Artificial Intelligence and Statistics. PMLR, 2023.

---

> ### Author Response · Authors · 2024-12-03
> **Gentle Reminder for Your Valuable Feedback**
>
> I hope this finds you well. This is a gentle reminder that there is less than one day remaining before the discussion period concludes. We would greatly appreciate any further comments or feedback you might have on our revisions to ensure all concerns are thoroughly addressed.
>
> Thanks for your time and considerations.
>
> Best regards,
>
> The Authors

---

### Official Review · Reviewer_G8sE · 2024-11-03

**Soundness:** 3
**Presentation:** 3
**Contribution:** 2
**Rating:** 5
**Confidence:** 4

**Summary:**

This paper analyzes the in-context decision-making abilities of LLMs in dueling bandits, which is a preference-based variant of the multi-armed bandit problem. The authors compare the performance of various model including GPT-4 and Llama in small dueling bandit problems (e.g., K=5 arms). Here, the performance of the LLMs is measured by their weak and strong Codorcet winner regret. Overall, the results suggest that the LLMs are incapable of realiably minimizing regret with the exception of GPT-4 Turbo (w.r.t. weak regret). The authors finally propose to combine LLM and explore-then-commit dueling bandit algorithms.

**Strengths:**

- I believe there is currently quite some interest in evaluations of in-context decision-making capabilities of LLMs w.r.t. exploratory and interactive learning behavior (see, e.g., prior work by Krishnamurthy et al. (2024)).
- The authors test a wide range of models including GPT-4 Turbo, Llama 3.1, and o1-preview.
- The experiments are described in detail.
- The authors propose a framework to integrate LLMs with existing (explore-then-commit) dueling bandit algorithms, which is conceptually interesting.

**Weaknesses:**

- All evaluations are performed on utility-based (i.e., reward-based) dueling bandits with a linear link function, which is arguably as similar as one can formulate the dueling bandit to the multi-armed bandit. This doesn't really capture the full complexity of learning from preference feedback. In particular, throughout all considered problems SST and STI are satisfied and no ablation w.r.t. the effect of this additional structure on in-context learning abilities are included. It would be much more interesting to test the ability of in-context learning on problems without transitivity in my opinion.
- Related to the above point, the paper only considers problem where the Condorcet winner exists. There are many alternative solution concepts in dueling bandits such as the Borda winner, Copeland winner, and von Neumann winner, which are guaranteed to exist.
- Oddly enough in your baselines you didn't include Versatile-DB (Saha and Gaillard 2022), which is the only near-optimal dueling bandit algorithm known (afaik) for the dueling bandit problem without additional assumptions.
- From my point of view the contributions of this paper are primarily empirical. Still, I want to comment on the shortcomings and lack of significance of the theoretical contributions:
	- Assumption 1 states that the LLM behaves the same as a uniform sampler in the worst-case so that Theorem 4.1 simply states that a uniform sampler has regret at least T/K in the worst-case, which is obvious. In Theorem 4.2, the properties of the LLM also don't matter since the guarantee for strong and weak regret simply uses that the number of additional comparisons due to the LLM is bounded by design (we don't even have to use the information generated by the LLM choices).

**Questions:**

- I don't understand the purpose of equation (4), i.e., the "relative decision window". Without defining the function $h$ I don't understand what it tells us. The function $I$ is also not defined concretely. Could you elaborate further? Is the point you're trying to make just that as the number of arms increases LLM performance decreases?
- The claim (line 224) that "This reveals relative decision-making abilities emerge as the general capabilities of LLMs grow via ..." is too strong in my opinion, as the evaluation are performed only on dueling bandits with a strict order. Could you defend this claim; especially, in view of the limited class (and size) of dueling bandit problems you consider in this paper?

Other minor suggestions:
- Make sure to use \citep instead of \cite or \citet to have parentheses around your references when they are not referred to actively: For example, in the first line, it should be (Wei et al. 2022) instead of Wei et al. (2022).
- Line 36 typo: rewards instead of awards

---

> ### Author Response · Authors · 2024-11-28
> **Rebuttal by Authors (Part I)**
>
> We thank reviewer G8sE for the insightful feedback, which has helped us clarify key details in the updated manuscript. Please see our responses addressing the specific concerns below:
>
> 1. > All evaluations are performed on utility-based (i.e., reward-based) dueling bandits with a linear link function, which is arguably as similar as one can formulate the dueling bandit to the multi-armed bandit. This doesn't really capture the full complexity of learning from preference feedback. In particular, throughout all considered problems SST and STI are satisfied and no ablation w.r.t. the effect of this additional structure on in-context learning abilities are included. It would be much more interesting to test the ability of in-context learning on problems without transitivity in my opinion.
>
> We really appreciate your valuable feedback on the transitivity. We have added related experiments for our top-performing LLM (GPT-4 Turbo) on the intransitive case (Intransitive-Easy and Intransitive-Hard). The environment section on Page 4 reorganizes the descriptions of environment instances, and the Scalability Limitation section on Page 6 includes an analysis of the experimental results. In summary, when removing transitivity in preference structures while keeping the number of arms the same, LLMs fail to replicate their exceptional weak regret performance.
>
> 2. > Related to the above point, the paper only considers problem where the Condorcet winner exists. There are many alternative solution concepts in dueling bandits such as the Borda winner, Copeland winner, and von Neumann winner, which are guaranteed to exist.
>
> Regarding the Winner setting, we want to clarify that the scope of our paper focuses on standard context-free dueling bandits (DB) with a Condorcet Winner [1]. We consider the following directions as important future work directions: (i) Integrating LLMs with other ongoing regret-minimization algorithms, complementing the explore-then-exploit methods discussed in our paper. (ii) Exploring LLMs’ performance under other winner definitions, such as Borda and Neumann winners. (iii) Investigating LLMs’ behavior in other DB settings, such as contextual dueling bandits, multi-dueling bandits, and adversarial dueling bandits.
>
> 3. > Oddly enough in your baselines you didn't include Versatile-DB (Saha and Gaillard 2022), which is the only near-optimal dueling bandit algorithm known (afaik) for the dueling bandit problem without additional assumptions.
>
> Thank you so much for pointing this out. We have added experiments with Versatile-DB to our baselines to ensure a more comprehensive and complete baseline comparison.
>
> 4. > In Theorem 4.2, the properties of the LLM also don't matter since the guarantee for strong and weak regret simply uses that the number of additional comparisons due to the LLM is bounded by design (we don't even have to use the information generated by the LLM choices).
>
> As for Theorem 4.2, we aim to clarify the **Best-of-Both-Worlds Design** of our agentic flow framework: the framework is explicitly designed to leverage the strengths of both standalone LLMs and robust DB algorithms. It benefits from the exploratory behavior of the LLM within the Relative Decision Boundary (RDB). The LLM's zero-shot reasoning capabilities allow it to prioritize promising arms during initial exploration, which reduces the exploration burden on the underlying DB algorithm.
>
> More specifically, we seek to utilize effective strategic behavior of LLMs while having an agent system capable of making robust decisions. Given that LLMs are inherently stochastic systems and can occasionally make errors due to various reasons like hallucinations or misunderstandings of tasks, they may fail in worst-case scenarios (as stated in Theorem 4.1). Therefore, in order to address this black-box nature, the framework is designed not to strictly depend on the information generated by the LLM. The bounded number of comparisons involving the LLM is a deliberate design choice to mitigate the inherent unpredictability of LLM outputs (e.g., hallucination or biased exploration) such that suboptimal LLM suggestions do not degrade the overall regret bound of the off-the-shelf algorithms. In summary, through the interplay between rule-based experts and in-context LLM agents, our framework sheds light on how language-based reasoning can inspire provably robust frameworks [2, 3] that translate words into actions.
>
> [1] Yue, Yisong, et al. "The k-armed dueling bandits problem." Journal of Computer and System Sciences 78.5 (2012): 1538-1556.
>
> [2] Dalrymple, David, et al. "Towards Guaranteed Safe AI: A Framework for Ensuring Robust and Reliable AI Systems." arXiv preprint arXiv:2405.06624 (2024).
>
> [3] Tegmark, Max, and Steve Omohundro. "Provably safe systems: the only path to controllable AGI." arXiv preprint arXiv:2309.01933 (2023).

---

> ### Author Response · Authors · 2024-11-28
> **Rebuttal by Authors (Part II)**
>
> 5. > I don't understand the purpose of equation (4), i.e., the "relative decision window". Without defining the function
>  I don't understand what it tells us. The function is also not defined concretely. Could you elaborate further? Is the point you're trying to make just that as the number of arms increases LLM performance decreases?
>
> Thanks for your insights on the transitivity issue, we found that the previous definition of the Relative Decision Window (mainly based on K) was incomplete. In the updated version, we revised the definition to Relative Decision Boundary (RDB) under the Scalability Limitation section on Page 6. The RDB is defined to describe the DB instances that LLMs can effectively handle (with a predefined acceptable level of weak regret). The form of the definition here refers to the definition of the CoT Reasoning Boundary (RB) definition in this NIPS oral paper [1]. Based on the black-box nature of LLMs, the way to quantify this generalization failure (LLMs' performance degrades when introducing intransitive preference structures or a large number of arms) is to provide an empirically grounded definition to capture this phenomenon under the influence of relevant factors.
>
> 6. > The claim (line 224) that "This reveals relative decision-making abilities emerge as the general capabilities of LLMs grow via ..." is too strong in my opinion, as the evaluation are performed only on dueling bandits with a strict order. Could you defend this claim; especially, in view of the limited class (and size) of dueling bandit problems you consider in this paper?
>
> The presentation here is indeed too strong. We have revised this claim to focus on our problem setup (see the first paragraph on page 5), specifically on the ability to solve the dueling bandit problem in-context (with both transitive and intransitive cases), rather than on relative decision-making.
>
> 7. > Make sure to use \citep instead of \cite or \citet to have parentheses around your references when they are not referred to actively: For example, in the first line, it should be (Wei et al. 2022) instead of Wei et al. (2022).
> Line 36 typo: rewards instead of awards
>
> Again, thank you so much for your careful review and extremely valuable feedback. We have made corresponding changes in the manuscript to incorporate all the points above.
>
> We hope this clarification of our work addresses the reviewer's concerns and provides a clearer understanding of our contributions. In light of this clarification, we hope you would consider increasing your score in support of this work. Otherwise, could you let us know what additional changes are necessary to make this work ready for publication?
>
> [1] Chen, Qiguang, et al. "Unlocking the Capabilities of Thought: A Reasoning Boundary Framework to Quantify and Optimize Chain-of-Thought." arXiv preprint arXiv:2410.05695 (2024).

---

> > ### Comment · Reviewer_G8sE · 2024-11-30
> >
> > Thank you for your response and the additional experiments you conducted; especially the ones on instances with intransitive preferences. I think this now provides a more complete and to some extent correct evaluation of the in-context decision-making capabilities of LLMs for preference feedback.
> >
> > I think my earlier apprehension was warranted as the results are very different for intransitive preferences, i.e., problems where the dueling bandit problem is very different from the multi-armed bandit problem. All LLMs seem to fail for intransitive preferences, which suggests that these models do indeed struggle to utilize general preference feedback (which should be mentioned in the abstract by the way). Overall, my impression is that the story/message of the paper ought to change because of this.
> >
> > As a result, I think that the paper would immensely benefit from another iteration / major revision and repositioning. This is the reason I want to keep my original score; even though I think that this is very promising work.

---

> ### Author Response · Authors · 2024-12-03
> **Rebuttal by Authors**
>
> Thank you for your insightful response! However, I believe the content I provided regarding the "fail to generalize" and "intransitive" parts may have misled you. We have further revised the manuscript to clearly position our conclusion. Here’s a clarification:
>
> For intransitive instances, while the LLM's weak regret may not exhibit exceptional performance across the entire time horizon as it does in transitive instances, Figures 10 and 11 (subfigures) demonstrate that the LLM's performance during the initial steps remains impressive, quickly including the Condorcet winner in duels. However, its inability to understand the intransitive structures leads to cycling comparisons, causing bad regret performance in the long run.
>
> We will add two figures of LEAD performance on intransitive instances and revise the presentation of abstract and conclusion according to your suggestions to convey the following key points:
>
> - The LLM’s linguistic prior allows it to quickly identify the Condorcet winner from the dueling history **(for both transitive case and intransitive case)**, but it is vulnerable.
>
> - The DB algorithm enables LEAD to maintain long-term performance theoretical guarantees.
>
> - We propose an agentic flow system that achieves best-of-both-worlds. Under our designed bounded match arm procedure, it effectively leverages the LLM’s exploration strength in its most capable regions while ensuring theoretically robust performance.
>
> Again, thank you so much for your insightful feedback to help us identify the lack of clarity in the manuscript and made corresponding revisions. Based on these additional clarifications, we hope you will consider increasing your score in support of this work. Otherwise, could you let us know what additional changes or improvements are necessary to make this work ready for publication?

---

### Author Response · Authors · 2024-11-28
**Summary of the Modifications Made in the Manuscript**

We sincerely appreciate the valuable feedback from the reviewers and will address them carefully in each individual rebuttal. We have revised the manuscript to address the concerns raised and to clarify the positioning and scope of our paper. We have highlighted the significant modifications in the current version as summarized below:

1. **Abstract**:
The abstract (and the overall presentation of the paper) has been reframed to clearly position our focus on the In-Context Reinforcement Learning (ICRL) [1, 2, 3, 4] capabilities of general-purpose LLMs and their challenges. We refined the presentation to emphasize that our method introduces an agentic flow framework to bridge the identified optimality gap between LLMs and DB algorithms.

2. **Introduction**:
The Introduction has been revised to emphasize ICRL's unexplored potential with LLM agents and to strengthen our motivation with a focus on understanding the emergent In-Context Dueling Bandit abilities and strategic behaviors of LLMs.

3. **Experimental Results**:
We added one baseline, Versatile Dueling Bandits [5], and two intransitive instances (Intransitive-Easy and Intransitive-Hard) to systematically evaluate LLM performance in the standard context-free Dueling Bandits setting with a Condorcet winner. We also adjusted the presentation to better summarize and explain LLM's success and failure modes.

4. **Conclusion**:
The conclusion has been revised to emphasize the limitations of LLMs, aiming to understand how language-based reasoning can inspire robust agentic frameworks that translate words into actions through the interplay between rule-based experts and in-context LLM agents.

5. **Appendix**:
We added exemplars of system biases, a comparison of o1 and 4turbo ratios (Best Arm Inclusion Ratio, Converged Best Arm Ratio, and Suboptimal Duel Ratio), a description of the Intransitive Case, and experimental results for Intransitive-Easy and Intransitive-Hard.

[1] Laskin, Michael, et al. "In-context reinforcement learning with algorithm distillation." arXiv preprint arXiv:2210.14215 (2022).

[2] Lee, Jonathan, et al. "Supervised pretraining can learn in-context reinforcement learning." Advances in Neural Information Processing Systems 36 (2024).

[3] Krishnamurthy, Akshay, et al. "Can large language models explore in-context?." arXiv preprint arXiv:2403.15371 (2024).

[4] Nie, Allen, et al. "EVOLvE: Evaluating and Optimizing LLMs For Exploration." arXiv preprint arXiv:2410.06238 (2024).

[5] Saha, Aadirupa, and Pierre Gaillard. "Versatile dueling bandits: Best-of-both-world analyses for online learning from preferences." arXiv preprint arXiv:2202.06694 (2022).

---

### Meta-Review · Area_Chair_4uZT · 2024-12-21

**Metareview:**

This paper evaluates LLMs as decision-makers in dueling bandits settings and proposes LEAD, a hybrid algorithm combining LLM capabilities with traditional dueling bandit algorithms. While showing some promise in weak regret performance with LLMs, the work has significant limitations including overly simplified evaluation settings, insufficient theoretical analysis, and unclear practical benefits.

**Additional Comments On Reviewer Discussion:**

During the discussion, reviewers raised several key concerns: (1) the unclear motivation for using LLM-guided dueling bandit algorithms without contextual information, (2) the incremental contributions of the LEAD framework compared to existing approaches, and (3) the reliance of theoretical results on overly restrictive assumptions that may not hold in practice. The authors responded by clarifying the scope, revising the manuscript for better presentation, adding experiments on intransitive preferences, and incorporating stronger baseline comparisons. While these efforts addressed some concerns, they did not fully resolve the concerns.

---

### Decision · Program_Chairs · 2025-01-22

Reject